# Improved biomass burning emissions from 1750 to 2010 using ice core records and inverse modeling

Bingqing Zhang [1], Nathan J. Chellman[2], Jed O. Kaplan[3], Loretta J. Mickley [4], Takamitsu Ito [1], Xuan Wang [5], Sophia M. Wensman[2], Drake McCrimmon [2], Jørgen Peder Steffensen [6], Joseph R. McConnell [2] & Pengfei Liu[1] ✉

Estimating fire emissions prior to the satellite era is challenging because observations are limited, leading to large uncertainties in the calculated aerosol climate forcing following the preindustrial era. This challenge further limits the ability of climate models to accurately project future climate change. Here, we reconstruct a gridded dataset of global biomass burning emissions from 1750 to 2010 using inverse analysis that leveraged a global array of 31 ice core records of black carbon deposition fluxes, two different historical emission inventories as a priori estimates, and emission-deposition sensitivities simulated by the atmospheric chemical transport model GEOS-Chem. The reconstructed emissions exhibit greater temporal variabilities which are more consistent with paleoclimate proxies. Our ice core constrained emissions reduced the uncertainties in simulated cloud condensation nuclei and aerosol radiative forcing associated with the discrepancy in preindustrial biomass burning emissions. The derived emissions can also be used in studies of ocean and terrestrial biogeochemistry.

Wildfires have an important effect on climate and atmospheric chemistry by emitting large amounts of greenhouse gases, reactive gases, and aerosols into the atmosphere[1]. Large uncertainties in historical fire emissions hinder understanding of fire-climate interactions, aerosol radiative forcing, and the accuracy of future climate change projections. In the present day (PD), remote sensing detections of active fire and burned areas provide constraints on global biomass burning (BB) emissions[2–5], although large regional discrepancies still exist among different inventories[6–8]. Historical BB emissions before the satellite era, which typically are constructed based on dynamic fire modeling and fire proxies, have even larger uncertainties in both magnitudes and temporal trends[9–11]. Some emission inventories, such as AeroCom[12] and BB4CMIP[13] show a relatively pristine atmospheric state in the preindustrial period (PI) compared to PD because they

assume a positive correlation between fire activities and population density. More recent studies suggest that these inventories may underestimate PI BB emissions because human activities may have reduced burned area by active fire suppression or more passively through landcover change and landscape fragmentation[14,15]. Comparisons with ice core fire proxies indicate that wildfire emissions during the late Holocene PI may be higher than those in PD[11,16].

Extensive efforts have been made to reconstruct historical BB emissions based on various types of proxy records, such as charcoal records[17,18], fire-scarred tree rings[19], and gases and aerosols preserved in ice cores[20–22]. Most of the studies, however, provide only qualitative temporal trends. Previous studies have demonstrated that both the trends and magnitude of global BB emissions can be constrained by ice core proxies of gases[20,23,24] using chemical transport model or box

[1]School of Earth and Atmospheric Sciences, Georgia Institute of Technology, Atlanta, GA, USA. [2]Division of Hydrologic Sciences, Desert Research Institute, Reno, NV, USA. [3]Department of Earth, Energy, and Environment, University of Calgary, Calgary, AB, Canada. [4]John A. Paulson School of Engineering and Applied Sciences, Harvard University, Cambridge, MA, USA. [5]School of Energy and Environment, City University of Hong Kong, Hong Kong SAR, China. [6]Physics of Ice, Climate, and Earth, Niels Bohr Institute, University of Copenhagen, Copenhagen, Denmark. ✉e-mail: pengfei.liu@eas.gatech.edu

model simulations, but the reconstructed emissions usually have relatively low temporal (>100 years) and spatial (hemispheric) resolutions, limited by the species lifetime and measurement resolution. Recent work by Eckhardt et al.[22] estimated the historical refractory black carbon (rBC) emissions in the Northern Hemisphere (NH) from 1850 to 2000 using inverse modeling constrained by an array of ice core records. The results highlighted a large discrepancy between the reconstructed rBC emissions and the a priori bottom-up estimates used in CMIP6, although the different sources of rBC (i.e., biomass burning vs. anthropogenic fossil fuel/biofuel) were not distinguished.

Quantitative estimates of global historical BB emissions that can be used as a boundary condition for climate modeling are scarce, representing one of the largest uncertainties in the assessment of aerosol climate forcing[11,16,25]. Previous studies have highlighted that BB emissions can largely shape the terrestrial cloud condensation nuclei baseline during the PI, and the uncertainty in PI BB emissions can be a major contributor to the overall uncertainty in the estimates of aerosol radiative forcing[16,25]. Accurate knowledge of PI-to-PD aerosol forcing also is essential for understanding the climate sensitivity of global temperature to changing greenhouse gases, as historical warming has been partially masked by aerosol cooling.

In this study, we use existing bottom-up historical emission inventories (BB4CMIP[13] or LPJ-LMfire[11,26] for BB emissions and CEDS[27] for fossil fuel/biofuel emissions) as the a priori emissions and combine these with a spatially extensive array of high-resolution, continuous measurements of ice-core rBC records developed from 31 sites in both hemispheres (Figure S1) as observational constraints. We reconstruct a posteriori BB emissions from 1750 to 2010 using an inverse model based on the chemical transport model GEOS-Chem. Our reconstructed BB emissions support previous findings that PI (represented by 1750–1780 in this study) BB emissions can be similar to or even exceed the PD level (represented by 1997–2010 in this study)[9,11,16], a scenario that is not currently considered in climate model studies. The reconstructed a posteriori emissions for several major BB source regions capture temporal variability not evident in a priori emissions and are generally consistent with regional climate proxy records[28,29], charcoal records[17,18], and ice core records of other BB tracers[30]. We further demonstrate that the ice-core rBC records provide observational constraints that can reduce the uncertainty of aerosol-cloud albedo forcing (CAF) associated with the uncertainty in the PI aerosol baseline. We also show that our gridded, model-ready a posteriori BB emissions can be used in Earth System Models to study the carbon cycle, marine biogeochemistry, and other climate-relevant endpoints.

## Results

### The reconstructed historical rBC emissions from 1750 to 2010

We divided the global rBC emissions into 17 continental regions following the divisions used in BB4CMIP[13] (Table 1, Figure S1). We derived the a posteriori BB emissions BB4CMIP_post (from a priori emissions of BB4CMIP[13]) and LPJ-LMfire_post (from the a priori emissions of LPJ-LMfire[11,26]) in each region for each year from 1750 to 2010, using inverse modeling based on ice core rBC records and deposition-emission sensitivities simulated by GEOS-Chem (Methods). Both inversions also use a priori anthropogenic fossil fuel and biofuel emissions from CEDS[27]. We set the a priori emission error by assuming that the emission uncertainties were larger in earlier years and gradually decreased towards the PD (Methods). Despite large discrepancies in the a priori emissions (Fig. 1a-b), the two a posteriori BB emissions show similar magnitudes and temporal trends, especially in earlier years when observational constraints were stronger (Fig. 1c-d). Both a posteriori BB emissions are relatively steady at ~2.1 Tg a⁻¹ rBC from 1750 until the early 1800s, higher than that in BB4CMIP (~1.6 Tg a⁻¹) and lower than that in LPJ-LMfire (~2.4 Tg a⁻¹); both a posteriori emissions decline at around 1850, followed by an increase until around 1930. This temporal trend is predominantly

governed by emission variations in South Hemisphere Africa (SHAF), which contributes ~30% of the global total BB emissions. From the late 1900s to the present, both a posteriori emissions tend to converge to the a priori emissions because we assumed that the a priori emissions in PD, which are constrained by satellite observations, have smaller errors than those in the early years (Methods). The two a posteriori emissions show similar PI-to-PD ratios, both of which are larger than 1. These results support previous findings that BB emissions in PI could be similar or even higher than in PD, based on ice core measurements[9,21,31], charcoal measurements[17], and fire modeling[11,16].

The a posteriori emissions are constrained by rBC flux measurements from a global array of ice core records. These ice core sites have different sensitivities to rBC emissions from specific source regions, therefore providing constraints on emissions at regional to hemispherical scales. However, due to the relatively short lifetime of rBC (~5 days in our simulations, consistent with other observed[32,33] and simulated results[34,35]), our records, which are mostly clustered in the polar regions, mainly reflect emission variations in high- and mid-latitude regions that are closer to the ice core sites and have a large contribution to the measured deposition fluxes. Specifically, rBC in Antarctic ice cores is mainly sourced from and sensitive to emissions in SHAF, South America (ARCD and SARC), and Australia (AUST), while rBC deposited in Greenland primarily originates in North America (BONAW, BONAE, TENAW, TENAE), Europe (EURO), and Boreal Asia (BOAS) (Figures S2-S3). Our analysis suggests that emissions in these regions (40% − 60% of the global total) are relatively well constrained by ice core data as indicated by similarities of two a posteriori emissions in this region after the inverse modeling process (Figure S4), compared with emissions from low-latitude regions, including North Hemisphere Africa (NHAF), North Hemisphere South America (NHSA), Central America (CEAM), Equatorial Asia (EQAS), and Middle East (MIDE), which have relatively small contributions to the deposition fluxes observed at the ice core sites (Figures S2-S3). As a result, a posteriori emissions in these regions are essentially unchanged from their a priori emissions (Figure S5). Even so, comparisons between the two a priori BB emissions indicate that high- and mid-latitude regions account for most of the discrepancies in global total emissions in PI, and our inverse modeling can provide a tight constraint for these regions and reduce discrepancies, further improving the estimates of the global total BB rBC emissions in the PI (Figure S6).

Regarding Greenland and Antarctica, our simulations using different emissions emphasize the contrast in relative importance between anthropogenic fossil fuel/biofuel and BB emissions across different regions (Fig. 2). In Antarctica, using the a posteriori BB emissions BB4CMIP_post and LPJ-LMfire_post results in simulated rBC deposition that is in better agreement with the ice core data compared with simulations using the a priori BB emissions, while using a posteriori anthropogenic fossil fuel/biofuel emissions does not provide further improvement. In contrast, simulations using a posteriori BB emissions in NH barely improve the consistency with ice core observations in Greenland, compared with those using the a priori BB emissions (Fig. 2b). In particular, the modeled results with the a priori fossil fuel/biofuel emissions fail to reproduce the rBC deposition peak between 1900 and 1950 that is evident in most Greenland ice cores and has been attributed to anthropogenic fossil fuel combustion from North America[36] (Figure S1). The a posteriori fossil fuel/biofuel emissions (i.e., CEDS_post, Figure S7) could improve the agreement between simulations and observations and suggest that fossil fuel/biofuel emissions in TENAE need to be 200% higher compared with the a priori CEDS estimate (~1.5 Tg a⁻¹ as suggested by the a posteriori CEDS emissions compared with ~0.5 Tg a⁻¹ in the a priori CEDS emissions, Figure S7) to agree with the observational constraints. This finding is consistent with a recent study of NH ice core rBC records that argued emissions in North America should be a factor of two higher than the

**Table 1 | The full names and abbreviations of 17 basic regions used in this study**

| No. | Abbreviation | Full name |
|---|---|---|
| 1 | ARCD | Arc of deforestation |
| 2 | AUST | Australia |
| 3 | BOAS | Boreal Asia |
| 4 | BONAE | Boreal North America – east |
| 5 | BONAW | Boreal North America – west |
| 6 | CEAM | Central America |
| 7 | CEAS | Central Asia |
| 8 | EQAS | Equatorial Asia |
| 9 | EURO | Europe |
| 10 | MIDE | Middle East |
| 11 | NHAF | Northern Hemisphere Africa |
| 12 | NHSA | Northern Hemisphere South America |
| 13 | SARC | South of the arc of deforestation |
| 14 | SEAS | Southeast Asia |
| 15 | SHAF | Southern Hemisphere Africa |
| 16 | TENAE | Temperate North America – east |
| 17 | TENAW | Temperate North America – west |

CMIP6 emissions (i.e., BB4CMIP + CEDS) for the period 1850-1920[22]. There are several ice core records sporadically located in other regions, including South America, North America, Boreal Asia, and Europe. Although these ice core records are not representative of the conditions in the whole region, using *a posteriori* BB emissions or anthropogenic fossil fuel/biofuel emissions could also result in a better agreement between the simulation results and ice core records (Figure S8).

## Temporal variations and comparisons with previous studies

In the SH, both *a posteriori* emissions consistently show BB rBC emissions decreased by ~40% from 1820 to 1860 and increased later until early the 1900s, while the two a priori emissions are stable during the same period. The increased temporal variability is constrained robustly by multiple ice core records measured at different locations in Antarctica (Figure S1), which are sensitive to emission changes in SH regions (i.e., ARCD, AUST, SARC, and SHAF). However, the high *a posteriori* error correlation in these regions indicates it is challenging to quantify the emissions in these regions independently (i.e., the same modeled results that match the observations could be obtained by adjusting emissions in either of the regions or a combination of several regions). Similar error correlation issues were discussed in previous work by Maasakkers et al.[37]. To show the results in the most extreme case, we further conducted sensitivity tests to examine how emissions in one of the four regions would change to match the observational values if emissions in all other three regions remain unchanged (Figure S9). These results provide an upper-limit estimate of emission changes constrained by ice core records. The sensitivity analysis suggests that adjusting emissions in ARCD or in SARC only marginally improves the agreement between the model and ice cores (Figure S9 k, l). Thus, this temporal variation is more likely driven by the emission trends in SHAF and AUST. While Figure S9 shows the upper limit of the *a posteriori* emissions in SHAF and AUST, the similar trends for both regions in the BB4CMIP$_{post}$ and LPJ-LMfire$_{post}$ (Figure S4) suggest that the simulated emissions are reasonable given that the ice core records at different locations of Antarctica consistently show similar trends as well. In addition, although emissions within different SH regions have uncertainties, total emissions in SH regions consistently show similar levels and trends (Figure S9).

We have identified a possible connection between the decrease in BB emissions in SHAF during the first half of the 19th century, as revealed in both *a posteriori* emissions and relatively sustained arid periods characterized by low precipitation. This hypothesis is supported by a comprehensive reconstructed "wetness" index in different middle and southern African regions based on documentary evidence and gauge data[38] (Figure S10b-d). This trend also is moderately correlated with the reconstructed Palmer Drought Severity Index (PDSI) from Last Millennium Reanalysis (LMR, r = 0.33-0.36)[39] (Figure S10e). In the savanna and woodland ecosystems that are prevalent in central and southern Africa, wildfire activity can be limited by herbaceous fuel availability in dry seasons[40,41]. Decreasing precipitation, suggested by the evidence above, would hinder the accumulation of herbaceous fuel, decreasing fuel availability and wildfire activity.

In NH regions close to the ice core sites, the *a posteriori* BB emissions generally exhibit higher levels compared with the a priori emissions (Figure S4). The largest discrepancy is observed in BOAS, which is one of the most important BB source regions in the NH. The two *a posteriori* emissions consistently show higher BB emissions in 1750-1780 and around 1850 and a decreasing trend in the late 1800s. This variability is not captured by the a priori emissions and is constrained mostly by the rBC records from Akademii Nauk and Flade Isblink, where the contribution of BB to rBC deposition from BOAS is larger than that from North America (Figure S2). The decrease after 1850 in boreal Asia also is reported by a previous emission reconstruction study with similar ice core records, and attributed to anthropogenic emission changes in Russia due to economic collapse during the October Revolution and World War I[22]. In contrast, we suggest that the decreasing trend, together with the two earlier periods of high rBC emissions, is likely related to BB emission changes because the trends are consistent with other BB tracers (i.e., para-hydroxybenzoic acid, p-HBA, and vanillic acid, VA) measured in the Akademii Nauk ice core (Figure S11)[30].

Human influence on BB gradually increased relative to the climate influence as population expand rapidly following the 19th century[42]. The *a posteriori* BB emissions show a declining trend starting from the late 1800s to the early 1900s in multiple regions, including SHAF, South America (ARCD and SARC), North America (BONAE, BONAW, TENAE, TENAW), and BOAS (Figure S4), although population and anthropogenic emissions continue to rise during this period. The declining trend in BB emissions is consistent with the charcoal records from the same regions[17] (Figure S10f, Figure S11d, Figures S12-13). The opposite trends in BB emissions and population change in this period suggest an overall negative effect of human activities on BB, likely caused by the rapid expansion of agricultural landscapes from natural landscapes that reduce biomass fuel load, landscape fragmentation resulting in reduced the fire spread, and land management to controll fires[14,43,44].

## Implications on cloud albedo forcing

Regional scale factors (i.e., *a posteriori* and a priori emission ratios) were derived for BB rBC from 1750 to 2010. These rBC scaling factors can be applied to other species co-emitted from BB, such as organic carbon (OC), under the assumption that emission factors used in a priori emissions remain applicable. These BB emissions (Figures S14-S15), together with CEDS emissions, were used in GEOS-Chem to simulate the PI-to-PD changes in aerosol concentration, cloud condensation nuclei (CCN) concentration, cloud droplet number concentration, and CAF (See Methods).

The results reveal that the two different a priori BB emissions lead to similar global mean CCN concentrations (Fig. 3) with large spatial discrepancies (Figure S16a-b). Here, the CCN concentrations were calculated, including aerosols from anthropogenic, biomass burning, and natural sources in both PI and PD. These spatial discrepancies mainly arise in SH regions due to large differences in BB emission levels

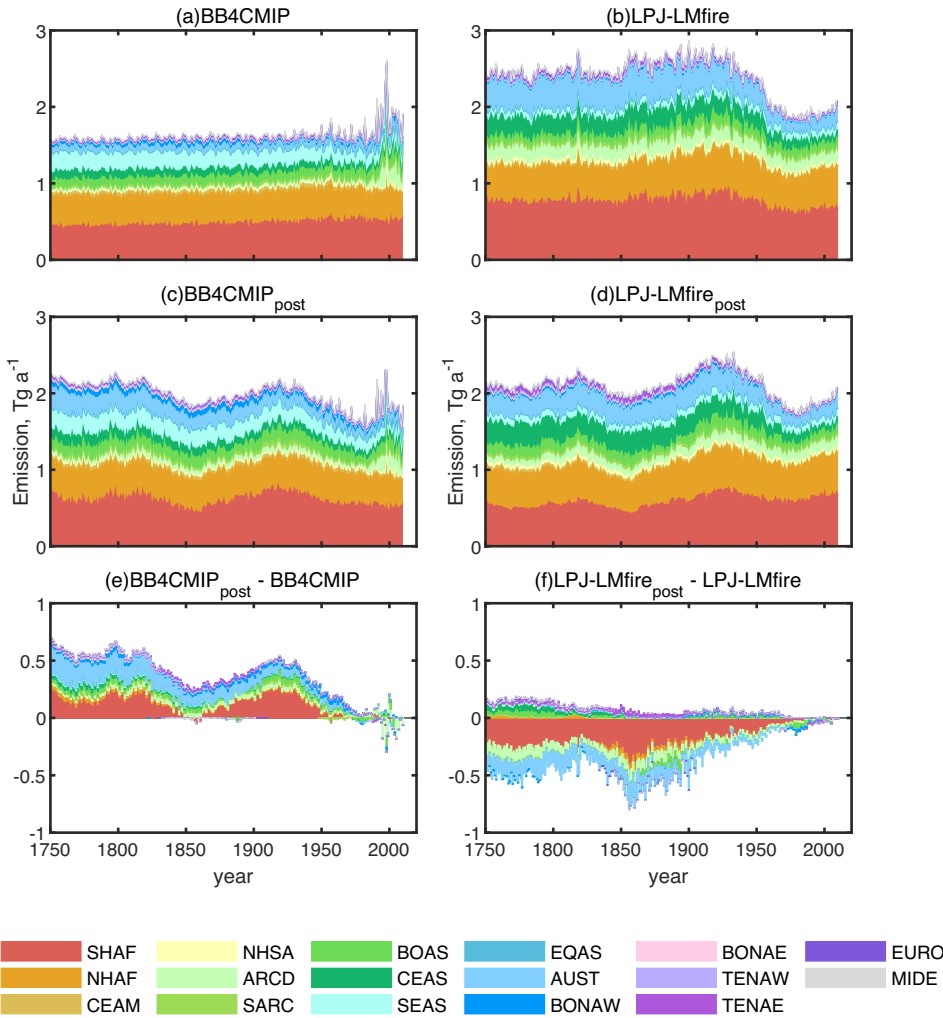

**Fig. 1 | Temporal trends of biomass burning (BB) refractory black carbon (rBC) emissions from 1750 to 2010. a, b** Temporal trends of a priori emissions by regions for the BB4CMIP and LPJ-LMfire inventories. **c, d** Temporal trends of *a posteriori* emissions by regions. **e, f** The difference between *a posteriori* emissions and a priori *emissions* in different regions. Definitions of region acronyms are given in Table 1.

and spatial distributions in these regions (Figure S14-S15). The *a posteriori* BB emissions constrained by ice core records indicate a reduction in the difference in PI BB rBC emissions between the two inventories from 0.86 Tg a⁻¹ to 0.07 Tg a⁻¹ globally (Fig. 3a) and from 0.80 Tg a⁻¹ to 0.10 Tg a⁻¹ in SH (Fig. 3c). As a result, the differences in simulated total PI CCN concentration also are reduced (Fig. 3d-f, Figure S15). We further calculated CAF between PI and PD due to anthropogenic aerosol emission changes using the different BB emissions (Fig. 3, Figure S17). The calculated CAF uncertainty related to PI BB emissions reduced from 0.08 W m⁻² (i.e., difference of simulations using the two a priori BB emissions) to 0.04 W m⁻² (using the two *a posteriori* emissions) globally (0.13 W m⁻² to 0.09 W m⁻² in SH and 0.02 W m⁻² to 0.00 W m⁻² in NH). These results highlight that observational constraints from ice core records can significantly reduce uncertainties in BB emissions by improving the estimates of PI CCN baseline and PI-to-PD CAF. Two *a posteriori* emissions scenarios consistently suggest higher BB aerosol emissions in PI compared with the CMIP6 emissions widely adopted in current climate models (i.e., BB4CMIP + CEDS). This conclusion remains robust even when considering the emission uncertainties (Figure S4). These findings suggest higher CCN concentrations during the PI which will result in less cooling effects of CAF when employed in climate models. The current climate model using CMIP6 emissions might need to be reevaluated, given the underestimation of BB emissions in PI. Studies have found

that even small uncertainty in simulated aerosol radiative forcing could lead to significant differences in climate model results, such as global surface temperature and sea ice area⁴⁵. The reduced uncertainties with observational constraints demonstrated in this study could possibly help with improving the climate model performance and climate sensitivity estimations, further improving future climate change projections.

### Implications for nutrient deposition from biomass burning

One major finding of our study is that PI BB emissions in SH regions were probably higher than indicated by the widely used BB4CMIP emissions. This implies that BB in the PI was a more important source of nutrients to terrestrial and marine ecosystems than previously estimated. To illustrate this point, we estimated one important nutrient species, phosphorus (P) deposition from BB based on simulated rBC deposition and compared it with estimated P deposition from mineral dust. We assumed a P:rBC ratio of 0.0029 for BB aerosols⁴⁶ with solubility of 15%⁴⁷ and P:dust ratio to be 0.0018 with solubility of 5%⁴⁷.

Although dust is the dominant source of P deposition globally, BB could be an important source regionally⁴⁷,⁴⁸, especially in South America, southern Africa, and the tropical Atlantic Ocean (Figure S18), contributing to more than 60% of the soluble P deposition from the atmosphere regionally. Our results suggest 0.18 Gg a⁻¹ higher annual

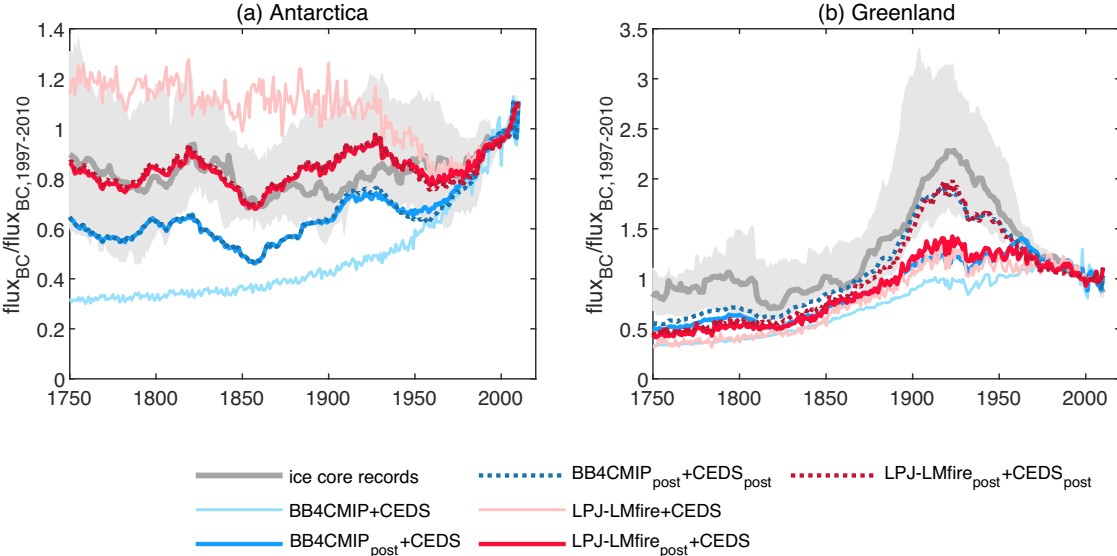

**Fig. 2 | Comparisons of measured and modeled trends of refractory black carbon (rBC) deposition fluxes. a** Comparisons with 14 Antarctica ice cores and (**b**) with 11 Greenland ice cores. The trends are shown as ratios of historical rBC flux to the present-day (PD) values (relative to the average value of the period from 1997 to 2010). The gray lines represent the median values of the measured ratios from ice core records, and the gray areas represent the 25th to 75th percentile range across sites. The modeled deposition flux is calculated using the product of the Jacobian matrix (**K**) and the emission vector (**x**) (see "Methods"). The two historical biomass-burning emissions inventories are BB4CMIP and LPJ-LMfire; the CEDS inventory includes fossil fuel and biofuel emissions.

soluble P emissions from BB in SH compared with baseline simulation using BB4CMIP, resulting in 0.06 Gg a⁻¹ higher soluble P deposition over land and 0.12 Gg a⁻¹ higher deposition over the ocean. The relative increases in annual P deposition were more than 50% in some regions compared with the baseline (Figure S18c). These increases are more pronounced during the boreal fall (September-November) when P deposition from dust is relatively minor (Figure S18 d). As a caveat, the above estimates only consider P emitted from BB and from mineral dust. Other sources also could be important. For example, biogenic sources are estimated to be as large as 10% to 100% of P emitted from mineral dust[48–50]. Our results, therefore, represent an upper-limit estimate.

Higher P deposition in the P-limited Amazon region could affect forest productivity and carbon uptake[51–53], while increased P in the N-limited tropical Atlantic Ocean could enhance nitrogen fixation by phytoplankton and further affect marine productivity and the global carbon cycle[54]. Our BB emissions from inverse modeling can affect the PI baseline state of the carbon cycle, and the associated climate effects could be further studied using coupled Earth system models.

## Discussion

Although our study utilized many of the same Arctic ice core rBC fluxes records and similar methods (inverse modeling with chemical transport model simulated sensitivities) with the previous study by Eckhardt et al.[22], here our aim is to provide a gridded, model-ready global-scale emission datasets that can be used directly in modeling studies. This is in contrast to the study by Eckhardt et al.[22] that mainly highlighted the large discrepancies between current widely used emission inventories and ice core data in the NH. Furthermore, we differentiated the *a posteriori* emissions between different sources (i.e., BB or anthropogenic fossil fuel/biofuel) and applied the gridded emissions in model simulations to estimate CAF and nutrient emissions and deposition. Consistency between results with two different a priori emissions and different meteorological conditions indicate the robustness of our results. Our reconstructed emissions capture the variability in historical BB that was not accounted for in previous inventories. However, there are some limitations to our approach. First, BB emissions in tropical and subtropical regions are poorly

constrained by polar ice core observations due to the short atmospheric lifetime of rBC and the long transport pathways to the poles. (The short rBC lifetime, on the other hand, can help differentiate emissions from different regions in high- and mid-latitude regions). Our reconstruction could thus be improved by incorporating ice core records of rBC in other regions, such as the Tibetan Plateau[55], or additional records from the Andes, which would help to constrain the large rBC emissions in Asia (CEAS and SEAS) or the Amazon. Second, we used rBC as a tracer because it is chemically stable and less susceptible to postdepositional alteration[11]. Further studies could benefit from incorporating multiple BB proxies with varying atmospheric lifetimes, such as ammonium, ethane, and carbon monoxide, to provide a more comprehensive understanding of both global and regional BB emissions. Third, our method can constrain emissions in only relatively large regions, but not on spatial scales within regions, which in the case of BB4CMIP and LPJ-LMfire differ greatly (Figures S14-S15). Finally, although this study primarily focuses on emissions over the past 250 years for which a priori emissions are available, future work could extend the study period to the Common Era (the past 2000 years) by incorporating longer ice core records, simulations of a priori emissions using fire models, and emission-deposition sensitivity simulated under different climate conditions.

## Methods
### Ice core measurements
The inverse modeling in this study is constrained by a total of 31 rBC records, including 29 previously published and unpublished datasets developed at the Desert Research Institute (DRI), as well as two additional published rBC records from other studies (Figure S1)[56]. Detailed information on the ice core sites can be found in Table S1. The DRI records presented here all were developed using the well-established Single Particle Soot Photometer (SP2) method as part of the DRI Continuous Flow Analytical (CFA) system[36,57], thus ensuring comparable uncertainties across the records. Annual deposition fluxes were determined using net water-equivalent snowfall rates derived from volcanically constrained, multi-parameter annual layer counting[58]. The rBC records were smoothed for the inversion process using a 30-yr running average to reduce large interannual variations due to climate

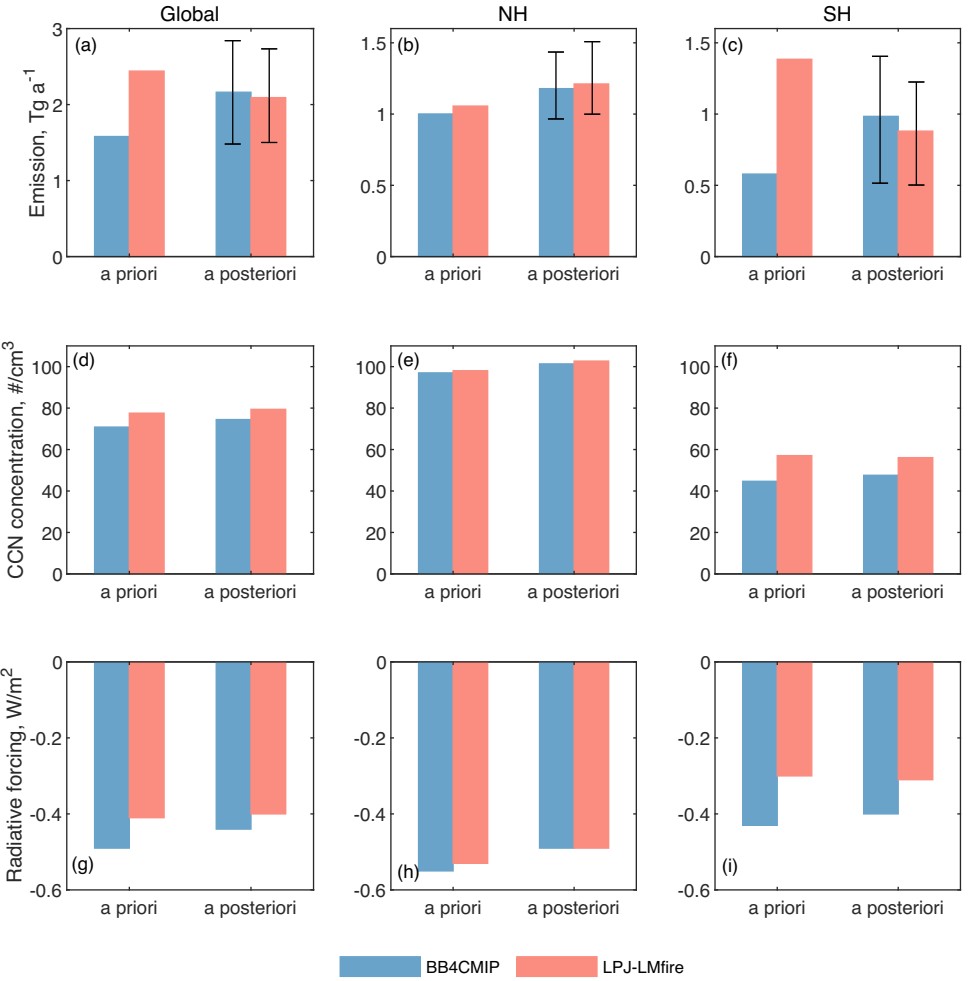

**Fig. 3 | Comparisons of biomass burning (BB) refractory black carbon (rBC) emissions, cloud condensation nuclei (CCN) concentrations, and cloud albedo forcing (CAF) using a priori and *a posteriori* BB emissions. a–c** Average rBC emissions during the preindustrial period (PI, 1750-1780) for (**a**) global, (**b**) North Hemisphere (NH), and (**c**) South Hemisphere (SH). The error bars represent the 2.5% and 97.5% range calculated by Monte Carlo simulations (Methods). **d–f** Average CCN concentration simulated by GEOS-Chem-TOMAS during PI (1750-1780) for global (**d**), NH (**e**), and SH (**f**). **g–i** Average CAF for PD (present day, 1997-2010) relative to PI (1750-1780) calculated by RRTMG radiative transfer model for global (**g**), NH (**h**), and SH (**i**).

variations, occasional fire events, and glaciological noise. As a caveat, data at some ice core sites are unavailable during the beginning of the study period. Any gaps in the rBC datasets were filled by assuming the same temporal trend of the nearest site with available data for the missing period. The error for such sections was assumed to be twice the error of the actual measurement.

## A priori emission inventories

Two BB emissions (i.e., BB4CMIP and LPJ-LMfire) and one fossil fuel/ biofuel emission (i.e., CEDS) were used in this study. BB4CMIP[13] is a historical BB emissions dataset starting from 1750 that is based on different records, including a satellite-based dataset (Global Fire Emission Database version 4 with small fires, GFED4s) after 1997, fire model results from the Fire Model Intercomparison Project (FireMIP) before the satellite era, and limited existing observational proxies (such as charcoal records and visibility observations) if available. LPJ-LMfire is a dynamic global vegetation model coupled with a fire module, with consideration of human influences on fire by hunter-gatherers, pastoralists, and farmers[26]. The model output is burned area and is scaled to the GFED4s burned area for the overlapping 1997-to-2015 period for each region[11]. Emission factors for different vegetation types were taken from GFED4s to derive emissions of rBC and other species. CEDS emission is a historical emissions dataset of

reactive gases and aerosols from anthropogenic fossil fuel and bio-fuel combustion[27]. BB4CMIP emissions are available at a spatial resolution of 0.25° × 0.25°, while LPJ-LMfire and CEDS emissions are available at a 0.5° × 0.5° resolution and are regridded in this study to 0.25° × 0.25°.

## GEOS-Chem simulations

We used GEOS-Chem v13.4.0 to construct the sensitivity matrix of rBC deposition flux to emission perturbations (Jacobian matrix) for the inversion analysis. The model has a horizontal resolution of 4° × 5° and 72 vertical layers and was driven by meteorology from the latest atmospheric reanalysis of the modern satellite era, MERRA2, from NASA's Global Modeling and Assimilation Office[59]. By default, the model assumed 80% of emitted rBC particles were hydrophobic (BCPO), with an e-folding time of 1.15 days before becoming hydro-philic (BCPI)[60]. We assumed fossil fuel/biofuel emissions were injected from the surface layer and BB emissions were injected partially below the planetary boundary layer (65%) and partially above the planetary boundary layer (35%)[61]. We ran the model simulations for five years, with emissions from 2000 to 2004 to represent the PD condition and with emissions from 1750 to 1754 to represent the PI condition. We used MERRA2 meteorology from 2000 to 2004 for both PI and PD scenarios, so the simulated differences between PI and PD were caused

solely by the emission difference, but we further analyzed the potential uncertainty related to different meteorology conditions using the meteorology input simulated by version E2.1 of the NASA Goddard Institute for Space Studies (GISS) general circulation model coupled in GEOS-Chem[62] (Text S1.1).

We assumed current emission inventories and meteorology conditions from MERRA2 in PD are accurate because they are all well-constrained by satellite data, energy use records, and direct measurements. We evaluated the model performance by comparing the simulated rBC deposition flux with the measured flux at the ice core sites between 2000 and 2004. The model shows relatively good performance, with a correlation coefficient ($r$) of 0.91. However, it overestimates the rBC flux by 11% to 12% (Figure S19 a, c). Since the coarse resolution model cannot accurately represent the high-altitude mountain regions where some of the ice core sites were located, we made corrections to the simulated wet deposition and dry deposition flux. Specifically, we corrected the simulated wet deposition by summing only the modeled flux in layers above the site altitude, and we corrected the simulated dry deposition by scaling with the ratio of rBC concentration in the layers of the site altitude to that at the surface. After these corrections, the slope of the regression line decreased from 1.11-1.12 to 1.03-1.02, indicating an improvement in model performance (Figure S19 b, d). We also conducted sensitivity tests for parameters that could affect the model performance, including fire plume injection height and rBC aging e-folding time. These tests show that changes in these parameters had limited effects on the simulated rBC deposition flux at the measurement sites, most of which are located far from the sources, and the results are consistent between different years, indicated by the relatively low standard deviations (Figures S20-S23).

Since rBC has a very low chemical reactivity in the atmosphere, with the major sources being BB and fossil fuel/biofuel emissions and sinks being wet and dry deposition[63], we assumed a linear relationship between rBC emission and deposition. To construct the Jacobian matrix, we tagged tracers of BCPI and BCPO from BB emissions as well as from anthropogenic fossil fuel/biofuel emissions in 17 regions. We calculated the sensitivity from each region and each source by dividing the total deposition fluxes at each grid cell by the total emissions of the corresponding tracers. Four Jacobian matrices with four sets of emissions are derived for the inverse modeling purpose (CEDS + BB4CMIP in 1750–1754, CEDS + BB4CMIP in 2000–2004, CEDS + LPJ-LMfire in 1750-1754, and CEDS + LPJ-LMfire in 2000-2004), and the average values in the 5-year periods were used as the final Jacobian matrices for each scenario. Although we treated emissions in each region as a whole, the different spatial patterns of emissions within each region between PI and PD could lead to large variations in the calculated sensitivities, most of which occur in the low-sensitivity conditions (Figure S24). To account for the effects of emission distribution transition from PI to PD in the inverse modeling process without consuming excessive computational resources to calculate the sensitivity matrices for the whole 261-year period, we simply used linear transition by assuming the sensitivity changed linearly from 1750 to 2000 and became constant after 2000. The details of the uncertainties are discussed in Text S1.1.

**Bayesian inversion framework**
We solved the *a posteriori* emissions of rBC from BB and fossil fuel/biofuel processes in each region ($\mathbf{x}$) using Bayesian inverse analysis with measured rBC deposition flux from ice cores ($\mathbf{y}$) as constraints. The cost function $\mathbf{J}$ of this problem can be described as:

$$J(\mathbf{x}) = (\mathbf{x} - \mathbf{x_A})^T \mathbf{S_A}^{-1} (\mathbf{x} - \mathbf{x_A}) + \gamma (\mathbf{y} - \mathbf{Kx})^T \mathbf{S_O}^{-1} (\mathbf{y} - \mathbf{Kx}) \qquad (1)$$

Where $\mathbf{x}$ is a 34 by 1 vector that represents the optimized estimates of BB emissions in 17 regions and fossil fuel/biofuel emissions in 17

regions, $\mathbf{x_A}$ is a 34 by 1 vector that represents the a priori BB emissions (from either BB4CMIP or LPJ-LMfire) and fossil fuel/biofuel emissions (from CEDS). Vector $\mathbf{y}$ with a size 31 by 1 is the observed rBC deposition flux from 31 ice core measurements. To overcome the model bias compared with the observations in PD (Figure S19), we scaled the observed values for the 2000–2004 period to the modeled values for the PD scenario to avoid unreasonable adjustments for the *a posteriori* emissions in the PD due to model bias instead of emission errors. We applied the same scale factors during the study period by assuming this model-observation bias does not change with time. Results without scaling show lower *a posteriori* emissions in the PD (Figure S25), contrary to our assumption that emissions in PD are accurate. The error covariance matrices for the a priori emissions ($\mathbf{S_A}$, with a size 34 by 34) and observations ($\mathbf{S_o}$, with a size 31 by 31) are represented by diagonal matrices, with the diagonal elements being the error variance of the emissions in each region ($\sigma^2_a$) and the error variance of the observations at each site ($\sigma^2_o$), respectively. All the off-diagonal elements are zero because we assumed the errors are uncorrelated with the neighboring values. Matrix K, with a size 31 by 34, is the Jacobian matrix. The constant $\gamma$ is a regularization factor to balance the a priori emission error (the first term of Eq. 1) and the observational error (the second term of Eq. 1).

The BB4CMIP emissions were constructed based on the ensemble of six fire models when no observational data were available[13]. We estimated the a priori error ($\sigma_a$) for BB4CMIP to be half of the range of the six fire model results and no less than 50% in each region. The resulting a priori emission errors for BB4CMIP range from 50% to 400%, similar to previous studies using errors 30% to 300%[6,22]. In the absence of better information, we set the a priori emission errors of LPJ-LMfire based on an initial guess (50%) and adjusted the error to avoid overfitting (i.e., negative *a posteriori* emission values) while maintaining consistency with observations. This leads to the a priori error ($\sigma_a$) to be 5% for SH regions (including ARCD, AUST, SARC, and SHAF), where simulations with LPJ-LMfire already show good agreement with observations in Antarctica[11]; 10% for NH regions with high emission levels (i.e., NHAF and CEAS) and 50% in all other regions. We estimated the a priori error for CEDS to be 100% between 1750-1850, with a linear decrease to 30% between 1850 and 2000 and remaining at 30% thereafter[22]. With these assumptions, the uncertainties of fossil fuel/biofuel emissions were generally smaller than those of BB emissions under most conditions. In the PI period, the fossil fuel/biofuel emissions were relatively low and began to significantly increase only after 1850. Even when we assumed a large relative uncertainty of 100% for fossil fuel/biofuel emissions, the absolute values were still lower than those of BB emission uncertainties. After 1850, while fossil fuel/biofuel emissions increased, the assumed relative uncertainty began to drop, leading to lower absolute errors for fossil fuel/biofuel emissions compared to BB emissions. This reflected our assumption that fossil fuel/biofuel emissions were relatively more accurate than BB emissions because more records (such as energy production and consumption data and population) were available as constraints.

The observational errors ($\sigma_o$) include measurement errors ($\sigma_{o1}$) and atmospheric modeling errors ($\sigma_{o2}$)[6]. We assumed the measurement error to be 10% of the measured values or 20% of the hypothetical values used to fill missing data sections and the modeling errors to be ($\mathbf{y}$-$\mathbf{K}$×$\mathbf{x_a}$), ranging from 2% to 173% of the measured values at different sites. We aggregated these errors by assuming $\sigma^2_o = (\sigma^2_{o1} + \sigma^2_{o2})$. We set the regularization factor $\gamma$ to be 0.5 based on the L-curve plot of squared observational and model errors (Figure S26).

We used the analytical solution for $\partial \mathbf{J}/\partial \mathbf{x} = 0$ to obtain the optimal estimates of the *a posteriori* emissions $\hat{\mathbf{x}}$ by Eq. 2. The calculations were performed annually from 1750 to 2010 using observations, a priori emissions, and sensitivity matrix for each year. We obtained two sets of

*a posteriori* emissions separately with different a priori emissions (i.e., BB4CMIP + CEDS and LPJ-LMfire + CEDS).

$$\hat{\mathbf{x}} = \mathbf{x_a} + \left(\gamma \mathbf{K}^T \mathbf{S_o}^{-1} \mathbf{K} + \mathbf{S_A}^{-1}\right)^{-1} \gamma \mathbf{K}^T \mathbf{S_o}^{-1} (\mathbf{y} - \mathbf{Kx}) \qquad (2)$$

We used Monte Carlo simulations to evaluate the uncertainty of *a posteriori* emissions associated with all the inverse modeling parameters, including observational error, a priori BB emission error, a priori anthropogenic fossil/fuel biofuel emission error, and simulated emission-deposition sensitivity error. Detailed discussion and description of error distributions can be found in Text S1. The overall uncertainties of the *a posteriori* emissions were aggregated by $\sigma^2 = \sum \sigma_i^2$ and represented by the 2.5% to 97.5% range shown by the shaded area in Figures S4-S5.

The annual *a posteriori* emission from the optimization process $\hat{\mathbf{x}}$ is a 34 by 1 vector with the first 17 elements to be the *a posteriori* BB emissions in 17 regions and the last 17 elements to be the *a posteriori* fossil fuel/biofuel emissions in 17 regions, all for rBC. We assumed the spatial distribution of burned area and emission factors of all the species to be the same as the a priori emissions so we derived the gridded *a posteriori* emission of rBC as well as other species by scaling the gridded a priori emissions by the factor $\hat{\mathbf{x}}_{BC}/\mathbf{x}_{a,BC}$ for each species in each region.

## Cloud albedo forcing calculation

We performed simulations using GEOS-Chem with TwO Moment Aerosol Sectional (TOMAS) online microphysics to simulate aerosol number, mass, and size distribution[64]. The TOMAS module has 15 size sections ranging from 3 nm to 10 μm for aerosol components of sulfate, organic carbon, BC, sea salt, and dust. We performed simulations under fixed meteorological conditions in 2000 with average emissions from 1750 to 1780 for PI conditions and from 1997 to 2010 for PD conditions. We performed simulations in PI conditions and PD conditions with four sets of emissions, i.e., CEDS + BB4CMIP, CEDS + BB4CMIP_post, CEDS + LPJ-LMfire, CEDS + LPJ-LMfire_post.

We calculated the cloud albedo effect at the top of the atmosphere (TOA) based on monthly size-resolved aerosol output from GEOS-Chem-TOMAS following the steps in our previous study[11]. In short, we used the sectional representation of cloud activation[65] to calculate aerosol droplet number concentrations (CDNC) for an updraft velocity of 0.3 m s⁻¹. We assumed a uniform cloud droplet radius of 10 μm for the PD conditions, and we calculated the effective radii of low- and mid-level clouds (up to 600 hpa) by taking the ratio of CDNC between PI conditions and PD conditions to the one-third power. The change in radiative flux at the TOA due to the change in cloud effective radius was calculated by the offline version of the Rapid Radiative Transfer Model for Global climate models (RRTMG). The difference in aerosol radiative effect between PD and PI was the aerosol radiative forcing caused by BB emissions. As a caveat, the radiative forcing calculated here did not consider the rapid adjustments of atmosphere and surface to aerosol perturbations, cloud lifetime effect, and the climate feedback of the updated fire emissions. These effects could be further studied using fully-coupled Earth system models.

## Data availability

All the ice core rBC records used in this study is deposited in Arctic Data Center at https://doi.org/10.18739/A2KH0F13W. The BB4CMIP emission can be download from: https://esgf-node.llnl.gov/projects/input4mips/; the CEDS emission can be downloaded from: https://data.pnnl.gov/dataset/CEDS-4-21-21; the emission data of BC from BB4CMIP_post and LPJ-LMfire_post is deposited in Harvard Dataverse at: https://doi.org/10.7910/DVN/KB0ESS; the emission data of other species is available upon request from corresponding author due to the

large size. Source data for generating all the figures are provided with this paper. Source data are provided with this paper.

## Code availability

The chemical transport model GEOS-Chem is open access at: https://github.com/geoschem/geos-chem; the code for version 13.4.0 used in this study is available at https://zenodo.org/records/6511970. A sample code for the inverse modeling process is provided in this paper.

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

## Acknowledgements

We thank the students and staff of the DRI Ice Core Laboratory for assistance on the ice core measurements, as well as the U.S. and other field teams. The B53 and B40 ice core samples were provided by the Alfred-Wegener-Institut, the James Ross Island and Gomez samples by the UKRI-NERC-British Antarctic Survey, and the Aurora Basin and W10 samples by the Australian Antarctic Division. Samples from the Flade Isblink and Hans Tausen cores were provided by the University of Copenhagen and from the Akademi Nauk core by the Alfred-Wegener-Institut. The ice core rBC records and chronologies were developed at DRI primarily with support from NSF grants, including 0538416, 0839093, 0909541, and 1023672. This specific research was funded by the Geosciences Directorate of the NSF under grants AGS-2102917, 2102918, 2117844, and AGS-1702814. P.L. and B.Z. also acknowledge partial support by the start-up funding from Georgia Tech.

## Author contributions

P.L., N.J.C., L.J.M., and J.R.M. designed the study. B.Z. performed the GEOS-Chem simulations and inversion and wrote the initial manuscript. J.O.K. performed the LMfire simulations and contributed to the LPJ-LMfire emissions. N.J.C., S.M.W., D.M., J.P.S., and J.R.M. contributed to the ice core datasets. X.W. helped with radiative forcing calculations. T.I. helped with nutrient deposition estimations. N.J.C., J.O.K., L.J.M., T.I., X.W., S.M.W., D.M., J.P.S., J.R.M., and P.L. commented on data analysis and contributed to the writing of the manuscript.

## Competing interests

The authors declare no competing interests.
