## [Peer Review File · Nature Communications]

Improved Biomass Burning Emissions from 1750 to 2010 using Ice Core Records and Inverse ModelingReviewer #1 (Remarks to the Author):

General comments:

The authors estimated global biomass burning emissions data from 1750-2010 using "top-down" inverse approach, using historical emissions inventory data and constraints on ice core records of refractory black carbon deposition fluxes. The study analyzed biomass burning emission historical trends based on the reconstructed inventory. The novelty of this study, especially the methodology shall be clarified compared to the previous study about inversion emission based on ice core record of BC. Moreover, the inversion results should be verified and be carefully analyzed. For example, how to prove the inversion result is biomass burning emission rather than other fossil fuel combustion sources, and how to prove the rationality of results based on the ice core records of rBC. The significance and the scientific impact of this study also need to be described in detail. The concerns mentioned below should be addressed. Several main comment:

(1) The same meteorological conditions were set throughout the simulations over the years especially when building the Emission-deposition sensitivities matrix. Moreover, the study made assumptions that are not supported by data when setting the priori inventory errors. These processes all lead to large uncertainties in the inversion of emissions. The reliability of results based on the ice core records of rBC need to be further verified. In addition, how to prove the inversion result is biomass burning emission rather than other fossil fuel combustion sources. BB is not the only source of the rBC.

(2) The study should provide scientific evidence for the findings by providing additional data or obtaining new data. For example, in the section "Implications on cloud albedo forcing", the conclusion "the cooling effect of aerosol radiative forcing from PI to PD may be overestimated in current climate models" have yet to be proven fully. Further evidence is needed to demonstrate whether this conclusion is valid considering the inventory error and simulation error.

(3) This study have created the biomass burning emissions inventory for a period. The significance for present/future climate assessments and pollution analysis, and the applicability of the methodology for the inversion of the BB emissions need to clarify in this study.

Specific comments (P represent Page X which was shown in the lower right corner of reviewer proof):

P2 line 58: What is the dividing line between preindustrial period(PI) and present period (PD), the beginning of the Industrial Revolution ?

P2 line 58: Did " these inventories" refer to the lists estimated using satellite monitoring data?

P3 line 70-75: The literature cited in the article utilized BC precipitation data recorded in ice cores and the idea of inversion to obtain historical emission inventories. The methodology innovation of this study need to be further explained.

P4 line 102: In this paragraph, the article analyzed that the inverse emission inventories are characterized by different discrepancies from the a priori emission inventories in the PI and PD periods. What is the reason for this different discrepancies? In addition, the boundary between the PI and PD periods in Figure 1 should be to clearly mark.

P5 line 131-136: MS showed that the constraints on the ice core data work differently in different regions, what is the reason? Why the constraints are not effective in low latitude regions? Is it related to the location of the observation site or the sedimentation fraction of the region? This part needs to be more clearly explained.

P5 line 137-140: The BB emission in low latitudes region were seemingly not well constrained by ice core data. It cannot be shown that the inverse model can provide a tight constraint on the total global BB emissions in the PI. Actually, The BB emission in low latitudes region and accurate inversion result should be pay more attention

P6 line141-143: Were the in the modeling of the different emission inventories evident only different in these two regions? Are the differences not reflected elsewhere? Why?

P6 line161-175: What is the significance of the sensitivity analysis of emissions by region to the model? What is the role of sensitivity analysis?

P7 line 196-200: What does the passage mean?

P7 line 201-202: The MS mentioned that the downward trend in emissions from the late 19th to early 20th centuries could be attributed to changes in the nature of human impacts on wildfires. The author should expand on why this change led to the decline in emissions.

P8 line 235-237: This result seems to be one of the important conclusions of this research and a new finding in reconstructing emission inventories relative to the priori inventories. Is it possible to further explain what can be further analyzed by obtaining this result? What are the new

implications of this result for the climate environment?

P8 line 246-249: The statement that "biomass burning is also an important regional source of global P deposition" needs to be supported by support data or other literature.

P10 line 294-296: The study should state that how to determine the processed rBC data were all from biomass burning and verify the reliability of the inversion result.

P11 line 342-344: All these tests mentioned in MS were conducted with the year 2000 as the scenario. Would the test situation vary in different years? The test scenarios for other years should be added to further supplement the cases to enhance persuasiveness.

P12 line 359-363: It is necessary to explain why the sensitivity is assumed to change linearly from 1750 to 2000 and remain constant after 2000.

P12 line 363-366: The study discussed the reasons for using the same meteorological conditions in the multi-year simulations in a Supplementary Document. However, it is still not sufficient to justify it because the results of the simulations under different meteorological conditions differed significantly in some areas, as illustrated in Figure S25.

P12 line 377-379: The reasons and justification for the adjustment of the measurements should be further clarified, and a detailed description of how the adjustments were made should be presented.

P13 line 388-408: It is necessary to further prove the reasonableness of the assumption of the range of the priori errors. The practical evidence of current assumption is not enough, and the uncertainty of the priori inventory can be determined by comparing the simulation using the priori inventory and the observation, or by conducting several experiments to set different error ranges to prove the reasonableness of the setting.

P14 line 433-434: It can be found that the uncertainty of emissions seems to be different in each region, and the uncertainty range of emissions in different regions should be marked in the form of a map.

P18 figure3: The sequence number of the subgraph should be indicated.

Reviewer #2 (Remarks to the Author):

Review of "Improved Estimates of Biomass Burning Emissions from 1750 to 2010 using Ice Core Records and Inverse Modeling"

This work uses inverse modeling to reconstruct biomass burning (BB) emissions spanning the preindustrial-to-present day period (1750 to 2010), using records of refractory black carbon (rBC) from 31 ice cores, two emission inventories (BB4CMIP and CEDS) and emission-deposition sensitivities simulated by the atmospheric chemical transport model GEOS-Chem. The authors found that preindustrial era BB emissions can match or exceed emissions from present day, a scenario not currently considered in climate model studies, and suggest that the cooling effect of aerosol radiative forcing from preindustrial era to present day may be overestimated in current climate models. The authors also found that preindustrial era BB emissions were higher than indicated by one of the emissions inventories used (BB4CMIP) for the southern hemisphere, suggesting that BB was a more important source of nutrients for terrestrial and marine ecosystems than previously estimated.

Considering the overall uncertainty related to BC in climate forcing estimates regionally and globally, specially in the preindustrial era, the results and conclusions presented in this work are of significance. This work is original, the methodology presented was sound and thoroughly detailed (both in the manuscript and supplementary information); this work used a robust dataset obtained by well established methods, and results support the conclusions claimed. I recommend this manuscript to be published with no additional comments or issues to resolve.

Manuscript ID: NCOMMS-23-36736-T

Title: Improved Biomass Burning Emissions from 1750 to 2010 using Ice Core Records and Inverse Modeling

Author(s): Bingqing Zhang, Nathan J. Chellman, Jed O. Kaplan, Loretta J. Mickley, Takamitsu Ito, Xuan Wang, Sophia M. Wensman, Drake McCrimmon, Jorgen Peder Steffensen, Joseph R. McConnell, and Pengfei Liu

Response to Reviewer #1

Comment 1

The authors estimated global biomass burning emissions data from 1750-2010 using "top-down" inverse approach, using historical emissions inventory data and constraints on ice core records of refractory black carbon deposition fluxes. The study analyzed biomass burning emission historical trends based on the reconstructed inventory. The novelty of this study, especially the methodology shall be clarified compared to the previous study about inversion emission based on ice core record of BC. Moreover, the inversion results should be verified and be carefully analyzed. For example, how to prove the inversion result is biomass burning emission rather than other fossil fuel combustion sources, and how to prove the rationality of results based on the ice core records of rBC. The significance and the scientific impact of this study also need to be described in detail.

Response

We thank the reviewer for their review and constructive comments. We conducted a more detailed analysis of the uncertainties related to the parameters within the inverse modeling process and discussed how these uncertainties may affect the results. We clarified the statements that could cause confusion pointed by the reviewer, including the novelty of this study compared with the previous work, PI and PD period definition, conclusions related to aerosol radiative forcing. We update the results, figures, discussions related to the changes throughout the manuscript. We hope that this revised version of the manuscript addresses all of the reviewer's concerns.

Comment 2

The same meteorological conditions were set throughout the simulations over the years especially when building the Emission-deposition sensitivities matrix. Moreover, the study made assumptions that are not supported by data when setting the priori inventory errors. These processes all lead to large uncertainties in the inversion of emissions. The reliability of results based on the ice core records of rBC need to be further verified. In addition, how to prove the inversion result is biomass burning emission rather than other fossil fuel combustion sources. BB is not the only source of the rBC.

Response

Meteorological conditions: We agree with the reviewer that the change in meteorological conditions from PI to PD can cause changes in emission deposition sensitivity. In the revised manuscript, we added new sensitivity tests using PI and PD emission deposition sensitivity matrices, constructed based on GISS-GC simulations using the CMIP6 historical meteorology from the GISS model (see Methods). The GISS meteorology shows large change from PI to PD¹ but has limited effects on the inverse modeling results (Fig. S30). The results indicate that our inversion remain robust under different meteorological conditions.

Figure S30 Timeseries of *a posteriori* BB rBC emissions with different emission deposition sensitivity matrices.

A priori inventory errors: We estimated the *a priori* emission errors of BB4CMIP based on the magnitude of the relative difference among the six fire models (Methods), assuming largest discrepancies among model results suggest larger uncertainties. The resulting *a priori* emission errors for BB4CMIP range from 50% to 400%, consistent with previous studies using errors 30% to 300%^{2,3}. We set the *a priori* emission error of LPJ-LMfire based on an initial guess (50%) and adjust the error to avoid overfitting (i.e., negative emission values) while maintaining a good consistency with observations. We believe the *a priori* emission errors used in this study are reasonable and could lead to reasonable results (Figure 2).

We revised the Methods section to include more details in determining the *a priori* emission errors and the references in Lines 399-410 as follows:

“We estimated the *a priori* error (σ_a) for BB4CMIP to be half of the range of the six fire model results and no less than 50% in each region. The resulting *a priori* emission errors for BB4CMIP range from 50% to 400%, similar to previous studies using errors 30% to 300%^{2,3}. In the absence of better information, we set the *a priori* emission errors of LPJ-LMfire based on an initial guess (50%) and adjusted the error to avoid overfitting (i.e., negative *a posteriori* emission values) while maintaining a good consistency with observations. This leads to the *a priori* error (σ_a) to be 5% for SH regions (including ARCD, AUST, SARC, and SHAF), where simulations with LPJ-LMfire already shows good agreement with observations in Antarctica; 10% for NH regions with high emission levels (i.e., NHAF

and CEAS) and 50% in all other regions. We estimated the *a priori* error for CEDS to be 100% between 1750-1850, with a linear decrease to 30% between 1850 and 2000 and remaining at 30% thereafter³.”

To study the uncertainty, we further perturbed the error within a range ($\pm 20\%$, Text S1) and shown the related uncertainties in Figure S27. We thus do not expect setting *a priori* emission errors in this range could lead to large uncertainties and change our conclusions.

Reliability of the results: We used Monte Carlo simulations to estimate the uncertainty of the *a posteriori* emissions associated with different input parameters including Jacobian matrices, *a priori* emission errors, and observational errors. To do so, we perturbed each parameter separately and aggregated the total error by $\sigma^2 = \sum \sigma_i^2$. The description of this process in the main text in Lines 432-438 are modified as follows:

“We used Monte Carlo simulations to evaluate the uncertainty of *a posteriori* emissions associated with all the inverse modeling parameters, including observational error, *a priori* BB emission error, *a priori* anthropogenic fossil/fuel biofuel emission error, and simulated emission-deposition sensitivity error. Detailed description on error distributions and discussion can be found in Text S1. The overall uncertainties of the *a posteriori* emissions were aggregated by $\sigma^2 = \sum \sigma_i^2$ and represented by the 2.5% to 97.5% range shown by the shaded area in Figures S4-S5.”

In the revised SI text, we made substantial edits and added detail descriptions of assumptions and considerations related to how uncertainty in each input parameter were set in our model, as follows:

“Text S1. Uncertainties related to inverse modeling parameters.

In this study, we used Monte Carlo simulations to evaluate the uncertainty related to parameters in inverse modeling including simulated Jacobian matrices, *a priori* BB emission errors, *a priori* anthropogenic fossil/fuel biofuel emission errors, and observational errors. We quantified the uncertainty related to each parameter individually by conducting 1,000 simulations, wherein each parameter was perturbed according to assumed error distributions. The overall error was aggregate by assuming $\sigma^2 = \sum \sigma_i^2$. Detailed assumptions and related uncertainties were discussed as follows.

Text S1.1 Uncertainties related to Jacobian matrices.

To calculate deposition-emission sensitivity at each ice core site, we treated emissions in each region as a whole. However, sensitivities to these regional emissions can vary with spatial variations within a region across different inventories, potentially resulting in discrepancies in the calculated Jacobian matrices throughout the study period (Figure S24). Additionally, interannual fluctuations in meteorological conditions can affect the calculated Jacobian matrices. To estimate the uncertainty of Jacobian matrices due to spatial variations in emissions from PI to PD as well as interannual variations of meteorological conditions, we assumed that sensitivities follow a bimodal normal distribution. The means and standard deviations of the two normal distributions were assumed to be the average values

and the standard deviations of the calculated sensitivities from 5-year GEOS-Chem simulations using the PI (i.e., 1750-1754) and PD (i.e., 2000-2004) emissions, respectively. The resulting uncertainties of global BB emissions are shown in Figure S27(a).

We used the same meteorology data from 2000-2004 from MERRA2 to drive GEOS-Chem and calculate Jacobian matrices for both PI and PD due to the lack of reliable observation-constrained meteorology in PI. To account for the changes in meteorological conditions from PI to PD, we conducted additional sensitivity tests using GEOS-Chem driven by meteorological fields from the NASA Goddard Institute for Space Studies E2.1 (GISS-E2.1)¹. We performed simulations from 1852 to 1856 for the PI scenario and from 2002 to 2006 for the PD scenario. The estimated Jacobian matrices with PI meteorology ($K_{PI_{met}}$) were derived by scaling the sensitivity matrices with PD meteorology from MERRA2 ($K_{PD_{met}, MERRA2}$) with the PI-to-PD ratio simulated with GISS meteorology, i.e., $K_{PI_{met}} = K_{PD_{met}, MERRA2} \times (K_{PI_{met}, GISS} / K_{PD_{met}, GISS})$.

Our results reveal that the relative difference in total rBC deposition fluxes between these two scenarios generally falls within $\pm 30\%$, indicating a relatively modest impact of meteorology on the annual average deposition fluxes (Figure S28). Although the relative difference in some model grids can reach as high as 100%, these effects are less likely to be discernible in our observational data since these grid cells are located far from the ice core sites. Similarly, relative differences in the emission-deposition sensitivities for most polar ice core sites fall within the range of $\pm 30\%$ (Figure S29), a range narrower than the sensitivity error we assumed as discussed above (mostly range from -85% to 90%). Large relative differences mainly occur in cases with low sensitivity values, which have limited effects on the inverse modeling results. This conclusion can be further supported by the inverse modeling results with the default $K_{PD_{met}, MERRA2}$ and the scaled $K_{PI_{met}}$ (Figure S30). Overall, we concluded that our method of using fixed PD meteorological conditions in all simulations is justified, and the resulting uncertainty should be smaller than that shown in Figure S27(a).

Text S1.2 Uncertainties related to *a priori* BB emission errors.

Uncertainties related to *a priori* BB emission errors could affect the inverse modeling results. Too small errors might undermine the constraint of the observational data, leading to *a posteriori* emissions nearly identical to the *a priori* emissions. Too large errors might weaken the constraint of the *a priori* emissions, leading to overfitting to the observational records and unreasonable *a posteriori* emissions. In Monte Carlo simulations, the *a priori* error for both BB emissions are assumed to be uniformly varied within $\pm 20\%$. The resulting uncertainties of global BB emissions are shown in Figure S26(b).

Text S1.3 Uncertainties related to *a priori* anthropogenic fossil fuel/biofuel emission error.

We assumed the *a priori* anthropogenic fossil fuel/biofuel emission error to be uniform distributed within a range of $\pm 20\%$ of the emissions. We do not expect the uncertainty of this parameter could introduce large uncertainties into our inverse modeling results, as shown in Figure S26(c).

Text S1.4 Uncertainties related to observational error.

We assumed the observational error to be uniformly distributed within a range of $\pm 20\%$ of the measurement values and larger than 0%, while the default observational errors range from 10% to 174% (see Methods). We do not expect the uncertainty of this parameter could lead to large uncertainties to our inverse modeling results, as shown in Figure S26(d).”

We added Figure S27 in SI text as follows,

Figure S27. Timeseries of estimated BB rBC emissions from 1750 to 2010 in BB4CMIP_{post} and LPJ-LMfire_{post} inventories. The shaded areas represent the 2.5% and 97.5% uncertainty range of the *a posteriori* emissions related to the uncertainty of (a) Jacobian matrices (Text S1.1) (b) *a priori* BB emission errors (Text S1.2) (c) *a priori* anthropogenic fossil fuel/biofuel emission errors (Text S1.3) (d) observational errors (Text S1.4).

Comment 3

The study should provide scientific evidence for the findings by providing additional data or obtaining new data. For example, in the section "Implications on cloud albedo forcing", the conclusion "the cooling effect of aerosol radiative forcing from PI to PD may be overestimated in current climate models" have yet to be proven fully. Further evidence is needed to demonstrate whether this conclusion is valid considering the inventory error and simulation error.

Response

We have revised the manuscript to highlight the major conclusion that using observational constraints can reduce the uncertainty of CAF associated with the uncertainty in PI biomass burning emissions. The discrepancies between the two *a posteriori* emissions decrease compared with the two *a priori* emissions, consequently reducing the uncertainties of CCN simulations and CAF calculations related to BB emissions. Our results show that using *a posteriori* BB emissions leads to a lower CAF compared with that using BB4CMIP emissions, which is widely used in climate models. This result is broadly consistent with previous studies^{4, 5}. However, here we would like to emphasize the effect on the uncertainty of CAF.

To clarify this point, we revised the text in Lines 220-226 as follows:

“Two *a posteriori* emissions scenarios consistently suggest higher BB aerosol emissions in PI compared with the CMIP6 emissions widely adopted in current climate models (i.e., BB4CMIP +CEDS). This conclusion remains robust even when considering the emission uncertainties (Figure S4). These findings suggest higher CCN concentrations during the PI which will result in less cooling effects of CAF when employed in climate models. Current climate model using CMIP6 emissions might need to be reevaluated given the underestimate of BB emissions in PI.”

Comment 4

This study have created the biomass burning emissions inventory for a period. The significance for present/future climate assessments and pollution analysis, and the applicability of the methodology for the inversion of the BB emissions need to clarify in this study.

Response

To discuss the significance of the past BB emissions derived in this study, we added the following sentences in Lines 226-231:

“Study have found that even small uncertainty in simulated aerosol radiative forcing could lead to significant differences in climate model results, such as global surface temperature and sea ice area⁶. And the reduced uncertainties with observational constraints demonstrated in this study could possibly help with improving the climate model performance and climate sensitivity estimations, further improve future climate change projections.”

For the applicability of the inversion to derive BB emissions, we think this method could derive reasonable *a posteriori* emissions with reliable past observational constraints and reasonable model simulations results. We believe that we could easily extend the study to the earlier period with longer ice core records, model-simulated *a priori* emissions, and emission deposition sensitivity simulated under different climate conditions, as we discussed in Lines 284-288 in the main text.

Comment 5

P2 line 58: What is the dividing line between preindustrial period(PI) and present period (PD), the beginning of the Industrial Revolution ?

Response

Preindustrial (PI) period denotes the era before the significant increase of human influence (i.e., fossil fuel/biofuel combustion) on the climate. Climate variabilities in PI are predominately governed by natural factors, thus it is a baseline to estimate the extent of human impact on climate change. Present day (PD) refers to the more recent period with substantial human impact on the climate. Different studies cited in this manuscript use different periods to represent PI and PD. Our work used the period 1750-1780 to represent PI and 1997-2010 to represent PD, which is specified in the Method section (Lines 450-452). To avoid confusion, we emphasized these specific timeframes in the main text (Lines 58-59) as follows,

“Our reconstructed BB emissions support previous findings that PI (represented by 1750-1780 in this study) BB emissions can be similar or even exceed the PD level (represented by 1997-2010 in this study)^{4,5,7}, a scenario that is not currently considered in climate model studies.”

Comment 6

P2 line 58: Did " these inventories" refer to the lists estimated using satellite monitoring data?

Response

These emissions refer to the aforementioned emissions AeroCom and BB4CMIP. The emission used in AeroCom and most of the emissions used in BB4CMIP assumed a positive correlation between fire activity and population density, leading to lower emissions in PI compared with PD. In contrast, some studies (Refs 14, 15 mentioned in the following sentence) show a more complex relationship between fire activity and population, and a decrease in fire activity may occur as population increases. We thus suggest emissions with a simple positive correlation between fire activity and population in AeroCom and BB4CMIP might underestimate emissions in PI.

To avoid confusion, we revised the sentence in Lines 23-26 as follows,

“Some emission inventories such as AeroCom⁸ and BB4CMIP⁹ show a relatively pristine atmospheric state in the preindustrial period (PI) compared to PD because they assume positive correlation between fire activities and population density.”

Comment 7

P3 line 70-75: The literature cited in the article utilized BC precipitation data recorded in ice cores and the idea of inversion to obtain historical emission inventories. The methodology innovation of this study need to be further explained.

Response

Our study utilizes similar data and methods with the study by Eckhardt et al³. Both studies use a chemical transport model to calculate emission-deposition sensitivity, use existing emission inventory as *a priori* emissions, and use ice core rBC deposition flux as observational constraints. However, this study has major improvement over the previous work.

First, the study by Eckhardt et al focuses on highlighting large discrepancies between current widely used emission inventories (i.e., CMIP 5 and CMIP 6) and ice core records. They provide an estimate of total emissions, but they did not differentiate between different sources (i.e., anthropogenic fuel combustions or BB), as we mentioned in Lines 42-43 in the manuscript. We focus more on correcting the current emission inventories based on the ice core records, and we aim to provide model-ready, gridded emission files that can be directly used in model simulations. We separated the emissions from anthropogenic combustion or BB, thus by comparing with other proxy records, we are able to identify that some discrepancies between observations and simulations can be explained by BB emissions (such as the temporal variations in SHAF and AUST before 1900s) and some are more likely explained by anthropogenic fuel combustion emissions (such as the rBC deposition peak in Greenland mentioned in). The study by Eckhardt et al provides an assumption that climate models might underestimate the BC radiative forcing in some regions and overestimate in other regions, while we are able to calculate radiative forcing with the gridded emissions derived in this study. We also provide another example of the use of our gridded emissions (i.e., phosphorus deposition). Our gridded emissions can also be used in other studies.

The study by Eckhardt et al focuses on NH only, and the study period started from 1850. We do the inversion on a global scale and investigate the results in both NH and SH, and we extend the study period from 1750 to PD, as pre-1800 conditions are widely used as the PI baseline in atmospheric radiative forcing studies^{4, 5, 10, 11}, and both of the *a priori* emissions (BB4CMIP and LPJ-LMfire) are available from 1750. By extending the spatial and temporal scope, we are able to identify and analyze more discrepancies between current emission inventories and observations. The *a posteriori* emissions derived in our studies bridge some of the gaps between observations and simulations (Figure 2a).

Third, the previous study uses the average values of CMIP5 and CMIP6 emissions as the *a priori* emissions. We use two *a priori* emissions which are significantly different from each other (Figure 1, a-b) and do the inversion separately. The similarity of two *a posteriori* emissions indicates the robustness of our results.

Last but not least, deposition flux at the ice core sites can be affected by both emissions and transportation (related to meteorology). While the previous study uses the same sensitivities during the study period (1850-2000), we conduct additional sensitivity test to prove the limited effects of meteorological condition change to the results (Figure S29).

To further clarify the differences between the previous study and our study, we add the following discussion in Lines 260-269,

“Although our study utilized many of the same Arctic ice core rBC fluxes records and similar methods (inverse modeling with chemical transport model simulated sensitivities) with the previous study by Eckhardt et al³, here our aim is to provide a gridded, model-ready global-scale emission datasets that can be used directly in modeling studies. This is in contrast to the study by Eckhardt et al (2023) that mainly highlighted the large discrepancies between current widely used emission inventories and ice core data in the NH. Furthermore, we differentiated the *a posteriori* emissions between different sources (i.e., BB or anthropogenic fossil fuel/biofuel) and applied the gridded emissions in model simulations to estimate CAF and nutrient emissions and deposition. Consistency between results with two different *a priori* emissions and different meteorological conditions indicate the robustness of our results.”

Comment 8

P4 line 102: In this paragraph, the article analyzed that the inverse emission inventories are characterized by different discrepancies from the a priori emission inventories in the PI and PD periods. What is the reason for this different discrepancies? In addition, the boundary between the PI and PD periods in Figure 1 should be to clearly mark.

Response

Larger *a priori* emission error (smaller S_A^{-1} in the cost function) in PI leads to smaller weights to the *a priori* emission constraint and larger weights to the observational constraints. Thus the two *a posteriori* emissions tend to converge to a similar level to match the observational records. On the contrary, smaller *a priori* emission error in PD leads to larger weights to the *a priori* emission constraints, thus the *a posteriori* emissions tend to converge with their *a priori* emissions.

We further clarified this point in the revised manuscript (Lines 78-80) as follows:

“We set the *a priori* emission error by assuming that the emission uncertainties are larger in earlier years and gradually decreased towards the PD (Methods). Despite large discrepancies in the *a priori* emissions (Figure 1 a-b), the two *a posteriori* BB emissions show similar magnitudes and temporal trends, especially in earlier years when observational constraints are stronger (Figure 1 c-d).”

There is no exact boundary for PI and PD periods and the definition varies with different studies. We added the definition of PI and PD period used in this study in the main text based on comments 5 for clarification.

Comment 9

P5 line 131-136: MS showed that the constraints on the ice core data work differently in different regions, what is the reason? Why the constraints are not effective in low latitude regions? Is it related to the location of the observation site or the sedimentation fraction of the region? This part needs to be more clearly explained.

Response

Different constraints in different regions are influenced by both the proximity of emission source regions to the observational sites and the relative contribution of emissions from these regions. When a region is closer to the observational site, the sensitivity of deposition to emission change in this region is higher; When emission in one region contribute more to the deposition at the ice core sites, then it is more likely that emission variations captured at this ice core site reflect emission change in this region.

To clarify this point, we revise the paragraph in Lines 95-109 as follows,

“However, due to the relatively short lifetime of rBC (~5 days in our simulations, consistent with other observed^{12, 13} and simulated results^{14, 15}), our records, which are mostly clustered in the polar regions, mainly reflect emission variations in high- and mid-latitude regions that are closer to the ice core sites and have a large contribution to the measured deposition fluxes. Specifically, rBC in Antarctic ice cores is mainly sourced from and sensitive to emissions in SHAF, South America (ARCD and SARC), and Australia (AUST), while rBC deposited in Greenland primarily originates in North America (BONAW, BONAE, TENAW, TENAE), Europe (EURO), and Boreal Asia (BOAS) (Figures S2-S3). Our analysis suggests that emissions in these regions (40% - 60% of the global total) are relatively well constrained by ice core data as indicated by similarities of two *a posteriori* emissions in this region after inverse modeling process (Figure S4), compared with emissions from low-latitude regions, including North Hemisphere Africa (NHAF), North Hemisphere South America (NHSA), Central America (CEAM), Equatorial Asia (EQAS), and Middle East (MIDE), which have relatively small contributions to the deposition fluxes observed at the ice core sites (Figures S2-S3).”

Comment 10

P5 line 137-140: The BB emission in low latitudes region were seemingly not well constrained by ice core data. It cannot be shown that the inverse model can provide a tight constraint on the total global BB emissions in the PI. Actually, The BB emission in low latitudes region and accurate inversion result should be pay more attention.

Response

Thanks for pointing this out. We would like to highlight that the inverse model can constrain emissions in middle- and high-latitude regions that have larger discrepancies compared with low latitude regions. Therefore, tight constraints on middle- and high-latitude regions can largely improve the estimate of total emissions. We also noted the limitation that ice cores have relatively weak constraints on low latitude emissions. This point has been discussed in the manuscript. To be more accurate, we revised the sentence in Lines 110-114 as follows,

“Even so, comparisons between the two *a priori* BB emissions indicates that high- and mid-latitude regions account for most of the discrepancies in global total emissions in PI, and our inverse modeling can provide a tight constraint for these regions and reduce discrepancies, further improve the estimates of the global total BB rBC emissions in the PI (Figure S6).”

Comment 11

P6 line141-143: Were the in the modeling of the different emission inventories evident only different in these two regions? Are the differences not reflected elsewhere? Why?

Response

We discussed the difference in Greenland and Antarctica because our ice core sites are mostly clustered in these two regions, providing data to compare and validate the model performance. The Greenland ice core records can represent the conditions in most of the NH regions, where anthropogenic emission is a more important rBC source especially after the industrial period, due to high population density and high anthropogenic fuel combustion. The Antarctic ice core records can represent the conditions in most of the SH, where anthropogenic emissions are relatively low due to lower population density and fuel combustion. The differences are also reflected in areas with one ice core record available (i.e., South America, Boreal Asia and Europe), and North America with three ice core records available. We added the comparison results in other regions in Figure S8, and added descriptions in the main text in Lines 133-137 as follows,

“There are several ice core records sporadically located in other regions including South America, North America, Boreal Asia, and Europe. Although these ice core records are not representative of the conditions in the whole region, using *a posteriori* BB emissions or anthropogenic fossil fuel/biofuel emissions could also result in a better agreement between the simulation results and ice core records (Figure S8).”

Comment 12

P6 line161-175: What is the significance of the sensitivity analysis of emissions by region to the model? What is the role of sensitivity analysis?

Response

To clarify this part, we revise the paragraph in Lines 144-160 as follows,

“However, the high *a posteriori* error correlation in these regions indicates it is challenging to quantify the emissions in these regions independently (i.e., the same modeled results that match the observations could be obtained by adjusting emissions in either of the regions or combination of several regions). Similar error correlation issues were discussed in a previous work by Maasackers et al¹⁶. To show the results in the most extreme case, we further conducted sensitivity tests to examine how emissions in one of the four regions would change to match the observational values if emissions in all other three regions remain unchanged (Figure S9). These results provide an upper limit estimate of emission changes constrained by ice core records. The sensitivity analysis suggests that adjusting emissions in ARCD or in SARC only can hardly improve the agreement between model and ice cores (Figure S9 k, l), thus this temporal variation is more likely driven by the emission trends in SHAF and AUST. While Figure S9 shows the upper limit of the *a posteriori* emissions in SHAF and AUST, the similar trends for both regions in the BB4CMIP_{post} and LPJ-LMfire_{post} (Figure S4) suggest that the simulated emissions are reasonable given that the ice core records at different locations of Antarctica consistently show similar trends as well. In addition, although emissions within different SH regions have uncertainties, total emissions in SH regions consistently show similar levels and trends (Figure S9).”

Comment 13

P7 line 196-200: What does the passage mean?

Response

In this paragraph, we found that the *a posteriori* emissions in BOAS show two periods of high BB emissions in 1750-1780 and in 1850, and a decreasing trend after 1850. The decreasing trend is also suggested by the previous study using Greenland ice core rBC records to reconstruct NH emissions³ (The consistency is mainly because we are using nearly the same ice core records in Greenland). But the previous study only estimate total rBC emissions and did not distinguish between anthropogenic fossil fuel/biofuel emissions and BB emissions. They attribute this trend (decrease in 1850 and increase afterwards) to anthropogenic emission change in BOAS caused by economic reasons (such as October Revolution and World War I). We suggest that this change is mainly related to changes in BB emissions, because ice core records of p-HBA and VA (BB tracers) shows similar trend at the same period. To clarify, we revise the paragraph in Lines 172-185 as follows,

“In NH regions close to the ice core sites, the *a posteriori* BB emissions generally exhibit higher levels compared with the *a priori* emissions (Figure S4). The largest discrepancy is observed in BOAS, which is one of the most important BB source regions in the NH. The two *a posteriori* emissions consistently show higher BB emissions in 1750-1780 and around 1850 and a decreasing trend in the late 1800s. This variability is not captured by the *a priori* emissions and is constrained mostly by the rBC records from Akademii Nauk and Flade Isblink, where the contribution of BB to rBC deposition from BOAS is larger than that from North America (Figure S2). The decrease after 1850 in boreal Asia also is reported by a previous emission reconstruction study with similar ice core records, and attributed to anthropogenic emission changes in Russia due to economic collapse during the October Revolution and World War I³. In contrast, we suggest that the decreasing trend, together with the two earlier periods of high rBC emissions, is likely related to BB emission changes because the trends are consistent with other BB tracers (i.e., para-hydroxybenzoic acid, p-HBA, and vanillic acid, VA) measured in the Akademii Nauk ice core (Figure S11)¹⁷.”

Comment 14

P7 line 201-202: The MS mentioned that the downward trend in emissions from the late 19th to early 20th centuries could be attributed to changes in the nature of human impacts on wildfires. The author should expand on why this change led to the decline in emissions.

Response

Agricultural expansion could have suppressed wildfire activity in the following aspect: Converting from natural landscapes (such as forest and grassland) to agricultural landscapes (land for cropping and grazing) could reduce the biomass fuel load; landscape fragmentation through roads, fence lines, paddocks, farmsteads, and other structures could suppress fire spread; and landscape management mainly aim to reduce uncontrolled fires. To clarify this, we revised the paragraph from Lines 186-196 as follows,

“Human influence on BB gradually increased relative to the climate influence as population expand rapidly following the 19th century¹⁸. The *a posteriori* BB emissions show a declining trend starting from the late 1800s to the early 1900s in multiple regions including SHAF, South America (ARCD and SARC), North America (BONAE, BONAW, TENAE, TENAW), and BOAS (Figure S4), although population and anthropogenic emissions continue to rise during this period. The declining trend in BB emissions is consistent with the charcoal records from the same regions¹⁹ (Figure S10f, Figure S11d, Figures S12-13). The opposite trends in BB emissions and population change in this period suggest an overall negative effect of human activities on BB, likely caused by rapid expansion of agricultural landscapes from natural landscapes that reduce biomass fuel load, landscape fragmentation resulting in reduced the fire spread, and land management to controll fires^{20, 21, 22}.”

Comment 15

P8 line 235-237: This result seems to be one of the important conclusions of this research and a new finding in reconstructing emission inventories relative to the priori inventories. Is it possible to further explain what can be further analyzed by obtaining this result? What are the new implications of this result for the climate environment?

Response

In this study we do not want to overly highlight this conclusion that we quantitatively estimate the aerosol cooling effect should be lower after using the revised PI aerosol emissions, while this has been discussed by several previous studies^{4, 5}. We revised this part in Lines 220-231 as shown in Comment 3 and Comment 4.

Comment 16

P8 line 246-249: The statement that "biomass burning is also an important regional source of global P deposition" needs to be supported by support data or other literature.

Response

We added two references in Line 241 from one observational study and one model study with a new compiled emission dataset. Our study with the current widely used emissions (i.e., BB4CMIP + CEDS, which is lower than other BB emissions used in this study) also suggests the importance of BB source to P deposition regionally (Figure S18). It is important to know that all these studies are subject to large uncertainties and further comprehensive evaluation will be needed to enhance our understanding of the process. However, such in-depth assessments fall outside of the scope of this study. We primarily aim to provide an example of the potential utility of our new emission datasets.

Comment 17

P10 line 294-296: The study should state that how to determine the processed rBC data were all from biomass burning and verify the reliability of the inversion result.

Response

We want to clarify that we did not assume all the measured rBC fluxes from the ice core records are from BB. Instead, we assumed that they are the sum of rBC emitted from BB and from anthropogenic fossil fuel/biofuel emissions. In the inverse modeling process, we used *a priori* emissions that include both BB emissions (i.e., BB4CMIP or LPJ-LMfire) and anthropogenic fossil fuel/biofuel emissions (i.e., CEDS), and we got the *a posteriori* emissions for both BB (Figure 1) and anthropogenic fossil

fuel/biofuel combustion (Figure S7). We assumed the current CEDS emission inventory is relatively accurate compared with BB emissions since more records (such as energy use, fuel consumption, and population data) are available as constraint, thus we set smaller *a priori* emission error for CEDS (see Methods). Although we did not further discuss the *a posteriori* CEDS anthropogenic fossil fuel/biofuel emission because we mainly focus on BB, we do include this source in our inverse model.

The measured rBC fluxes have high interannual variability which might be affected by many factors other than emission change, such as climate variations, occasional fire events and glaciological noise. We smooth the data to eliminate the other factors so that the changes in the smoothed records are most likely reflect the change of emissions.

Comment 18

P11 line 342-344: All these tests mentioned in MS were conducted with the year 2000 as the scenario. Would the test situation vary in different years? The test scenarios for other years should be added to further supplement the cases to enhance persuasiveness.

Response

To show that this conclusion is consistent in different years, we extended the simulation period for 5 years from 2000 to 2004. Instead of showing the relative difference in 2000, we now show the relative difference calculated based on the five-year averages (Figure S20 c and Figure S22 c), and we also show the standard deviation of the relative difference for the five years (Figure S20 d and Figure S22 d). There is no significant difference in the results when using the data for the year 2000 or the average values from 2000 to 2004. The standard deviation of the relative difference for the five-year period is mostly less than 5%. These further prove the consistency of the results in different years.

We revised the sentence in Lines 353-356 as follows,

“These tests show that changes in these parameters had limited effects on the simulated rBC deposition flux at the measurement sites, most of which are located far from the sources. The results are consistent between different years indicated by the low standard deviations (Figures S20-S23).”

We update Figures S20-S23 and the corresponding caption as follows,

“

Figure S20 rBC deposition fluxes in 2000-2004 simulated by GEOS-Chem with different fire emission injection heights. **a**, Modeled fluxes with default setting, which specifies that 65% of wildfire emissions are emitted within the boundary layer²³. **b**, Modeled fluxes from simulations in which 100% of wildfire emissions are emitted within the boundary layer. **c**, The relative differences between **(b)** and **(a)**, represent by (perturbed case – base case)/base case. The relative differences at the ice core sites are mostly less than 10%, indicating that plume heights might not have significant effects on the results in our study. **d**, The standard deviations of the relative differences in the five-year period. The standard deviations for the five-year period are mostly less than 5%, suggesting the consistency of the conclusion in different years.

Figure S21. Scatter plot showing rBC deposition fluxes at ice core sites for the base case versus the perturbed case with regard to fire injection heights. The settings of the base case and the perturbed case are described in Figure S19. Each point represents the average values from 2000 to 2004.

Figure S22 rBC deposition fluxes in 2000-2004 simulated by GEOS-Chem with different *e*-folding times for the rBC hydrophobic-hydrophilic conversion. **a**, Model outputs with the default *e*-folding time of 1.15 days. **b**, Model outputs with a shorter *e*-folding time of 8 hours, which could occur in heavily polluted regions²⁴. **c**, The relative differences between results in panels (**b**) and (**a**), represent by (perturbed case – base case)/base case. These results suggest a shorter *e*-folding time could lead to larger deposition near the source and smaller deposition in remote regions, including most ice core sites. Since such a short *e*-folding time is an extreme case in polluted regions, the results in panel (**c**) represents the upper limit of the deposition change with different *e*-folding times. **d**, The standard deviations of the relative differences in the five-year period. The standard deviations for the five-year period are mostly less than 5%, suggesting the consistency of the conclusion in different years.

Figure S23 Scatter plot showing BC deposition fluxes at ice core sites for the base case versus the perturbed case, with regard to hydrophobic-hydrophilic conversion. The settings in the base case and the perturbed case are as described in Figure S21. Each point represents the average values from 2000 to 2004.”

Comment 19

P12 line 359-363: It is necessary to explain why the sensitivity is assumed to change linearly from 1750 to 2000 and remain constant after 2000.

Response

In this study, to calculate deposition-emission sensitivity at each ice core site, we treat emissions in each region as a whole. However, sensitivity could change with different emission spatial variation within one region from PI to PD, potentially resulting in large discrepancies in the calculated Jacobian matrices over the study period (Figure S24). To account for the effects of emission distribution transition from PI to PD in the inverse modeling process without consuming too much computational resources on calculating the sensitivity matrices for the whole 261-year period, we simply assume the sensitivity changed linearly from 1750 to 2000 and became constant after 2000. We expect the resulting uncertainties to be within the range shown in Figure S27, which considers both the PI to PD emission spatial variation change and meteorological conditions change.

We revise the description in the Method Section in Lines 369-374 as follows,

“To account for the effects of emission distribution transition from PI to PD in the inverse modeling process without consuming excessive computational resources to calculate the sensitivity matrices for

the whole 261-year period, we simply used linear transition by assuming the sensitivity changed linearly from 1750 to 2000 and became constant after 2000. The details of the uncertainties are discussed in Text S1.1.”

Comment 20

P12 line 363-366: The study discussed the reasons for using the same meteorological conditions in the multi-year simulations in a Supplementary Document. However, it is still not sufficient to justify it because the results of the simulations under different meteorological conditions differed significantly in some areas, as illustrated in Figure S25.

Response

We revised the uncertainty estimate caused by different meteorological conditions based on Comment 2 and shown in Text S1.1. Additionally, we conducted sensitivity studies with K_{PD} as well as K_{PI} . The results suggest relatively small effects (Figure S30).

Comment 21

P12 line 377-379: The reasons and justification for the adjustment of the measurements should be further clarified, and a detailed description of how the adjustments were made should be presented.

Response

In the inverse modeling process, we assumed the PD emissions are accurate, and the simulated rBC deposition fluxes are accurate. Thus there should be little adjustment on the PD emissions for model results to match the observations. However, there is still bias between the simulated values and the observed values (Figure S19). Simply ignoring this bias will cause unreasonable adjustments on PD emissions, further causing error when estimating *a posteriori* emissions during the whole study period. We eliminated this error by scaling the observed fluxes to the modeled fluxes in PD. We assumed this bias between observations and simulations does not change during the study period. Thus for each year during the study period, we corrected modeled values with the same ratio in PD.

We revise the text in Lines 384-390 as follows,

“To overcome the model bias compared with the observations in PD (Figure S19), we scaled the observed values for the 2000-2004 period to the modeled values for the PD scenario to avoid unreasonable adjustments for the *a posteriori* emissions in the PD due to model bias instead of emission errors. We applied the same scale factors during the study period by assuming this model-observation bias does not change with time. Results without scaling show lower *a posteriori* emissions in the PD (Figure S25), contrary to our assumption that emissions in PD are accurate.”

Comment 22

P13 line 388-408: It is necessary to further prove the reasonableness of the assumption of the range of the priori errors. The practical evidence of current assumption is not enough, and the uncertainty of the priori inventory can be determined by comparing the simulation using the priori inventory and the observation, or by conducting several experiments to set different error ranges to prove the reasonableness of the setting.

Response

We revised the uncertainty estimate part related to *a priori* emissions error based on Comment 2 and shown in Text S1.2 (*a priori* BB emission error) and Text S1.3 (*a priori* CEDS emission error). We used Monte Carlo simulations by perturbing the *a priori* emission error in the range of $\pm 20\%$. The results in Figure S27 b-c show the related uncertainties.

Comment 23

P14 line 433-434: It can be found that the uncertainty of emissions seems to be different in each region, and the uncertainty range of emissions in different regions should be marked in the form of a map.

Response

The uncertainty of *a posteriori* emissions varied in different regions and in different years, and we marked the 2.5% to 97.5% uncertainty range in shaded areas in each panel in Figures S4-S5. The uncertainties shown here only represent uncertainties caused by our assumptions in different parameters (observational error, *a priori* BB emission error, *a priori* anthropogenic fossil/fuel biofuel emission error, and simulated emission-deposition sensitivity error). The small variations in the low latitude regions in Figure S5 does not mean emissions in these regions are accurate. Instead, this means our assumptions have little effects on emissions in these regions.

Comment 24

P18 figure3: The sequence number of the subgraph should be indicated.

Response

Done.

Reference

1. Murray LT, Leibensperger EM, Orbe C, Mickley LJ, Sulprizio M. GCAP 2.0: a global 3-D chemical-transport model framework for past, present, and future climate scenarios. *Geosci Model Dev* **14**, 5789-5823 (2021).
2. van der Velde IR, *et al.* Vast CO₂ release from Australian fires in 2019–2020 constrained by satellite. *Nature* **597**, 366-369 (2021).
3. Eckhardt S, *et al.* Revised historical Northern Hemisphere black carbon emissions based on inverse modeling of ice core records. *Nature Communications* **14**, 271 (2023).
4. Liu P, *et al.* Improved estimates of preindustrial biomass burning reduce the magnitude of aerosol climate forcing in the Southern Hemisphere. *Science Advances* **7**, eabc1379 (2021).
5. Hamilton DS, *et al.* Reassessment of pre-industrial fire emissions strongly affects anthropogenic aerosol forcing. *Nature Communications* **9**, 3182 (2018).
6. Fyfe JC, Kharin VV, Santer BD, Cole JNS, Gillett NP. Significant impact of forcing uncertainty in a large ensemble of climate model simulations. *Proceedings of the National Academy of Sciences* **118**, e2016549118 (2021).
7. van der Werf GR, Peters W, van Leeuwen TT, Giglio L. What could have caused pre-industrial biomass burning emissions to exceed current rates? *Clim Past* **9**, 289-306 (2013).
8. Dentener F, *et al.* Emissions of primary aerosol and precursor gases in the years 2000 and 1750 prescribed data-sets for AeroCom. *Atmos Chem Phys* **6**, 4321-4344 (2006).
9. van Marle MJE, *et al.* Historic global biomass burning emissions for CMIP6 (BB4CMIP) based on merging satellite observations with proxies and fire models (1750–2015). *Geosci Model Dev* **10**, 3329-3357 (2017).
10. Gordon H, *et al.* Reduced anthropogenic aerosol radiative forcing caused by biogenic new particle formation. *Proceedings of the National Academy of Sciences* **113**, 12053-12058 (2016).
11. Lee LA, Reddington CL, Carslaw KS. On the relationship between aerosol model uncertainty and radiative forcing uncertainty. *Proceedings of the National Academy of Sciences* **113**, 5820-5827 (2016).
12. Cape JN, Coyle M, Dumitrean P. The atmospheric lifetime of black carbon. *Atmospheric Environment* **59**, 256-263 (2012).
13. Lund MT, *et al.* Short Black Carbon lifetime inferred from a global set of aircraft observations. *npj Climate and Atmospheric Science* **1**, 31 (2018).
14. Hodnebrog Ø, Myhre G, Samset BH. How shorter black carbon lifetime alters its climate effect. *Nature Communications* **5**, 5065 (2014).

15. Samset BH, *et al.* Modelled black carbon radiative forcing and atmospheric lifetime in AeroCom Phase II constrained by aircraft observations. *Atmos Chem Phys* **14**, 12465-12477 (2014).
16. Maasakkers JD, *et al.* Global distribution of methane emissions, emission trends, and OH concentrations and trends inferred from an inversion of GOSAT satellite data for 2010–2015. *Atmos Chem Phys* **19**, 7859-7881 (2019).
17. Grieman MM, *et al.* Aromatic acids in a Eurasian Arctic ice core: a 2600-year proxy record of biomass burning. *Clim Past* **13**, 395-410 (2017).
18. Pechony O, Shindell DT. Driving forces of global wildfires over the past millennium and the forthcoming century. *Proceedings of the National Academy of Sciences* **107**, 19167-19170 (2010).
19. Marlon JR, *et al.* Climate and human influences on global biomass burning over the past two millennia. *Nature Geoscience* **1**, 697-702 (2008).
20. Andela N, *et al.* A human-driven decline in global burned area. *Science* **356**, 1356-1362 (2017).
21. Archibald S, Roy DP, Van Wilgen BW, Scholes RJ. What limits fire? An examination of drivers of burnt area in Southern Africa. *Global Change Biology* **15**, 613-630 (2009).
22. Andela N, van der Werf GR. Recent trends in African fires driven by cropland expansion and El Niño to La Niña transition. *Nature Climate Change* **4**, 791-795 (2014).
23. Val Martin M, Logan JA, Kahn RA, Leung FY, Nelson DL, Diner DJ. Smoke injection heights from fires in North America: analysis of 5 years of satellite observations. *Atmos Chem Phys* **10**, 1491-1510 (2010).
24. He C, Li Q, Liou KN, Qi L, Tao S, Schwarz JP. Microphysics-based black carbon aging in a global CTM: constraints from HIPPO observations and implications for global black carbon budget. *Atmos Chem Phys* **16**, 3077-3098 (2016).

Reviewer #1 (Remarks to the Author):

The review of the manuscript "Improved Biomass Burning Emissions from 1750 to 2010 using Ice Core Records and Inverse Modeling"

The revised manuscript demonstrates the significance of the study more clearly and several analysis of the results has been supplemented. However, there are still some issues in the manuscript that require further explanation. The MS can be considered for publication if these issues could be solved.

1. The study mainly utilized inversion techniques to provide the BC emissions. It is recommended to modify the title "...emission" to "...emission of BC".
2. According to the response and the associated revision, the study could not provide the improved emissions at low latitudes. The study needs to provide further quantitative data to demonstrate that the emissions at mid- and high-latitudes account for the majority of global emissions. Otherwise, it does not show that the results of this study improve global emissions. The title should be revised to be consistent with the research results (e.g., Improved... emissions at mid- and high-latitudes from...).
3. A priori inventories from different sources (e.g., biomass burning, fossil fuel/biofuel) were combined in the treatment of a priori inventories for inversion. However, the methodology described in the study could only obtain total rBC emissions and could not be able to distinguish the different emission sources of rBC, if the relationship between the different emission sources and the ice-core rBC records were not established separately. How did the study determine the emissions from Biomass Burning of rBC? Otherwise, it is recommended to modify the title to "Improved Emissions of BC at mid- and high-latitudes from...".

Manuscript ID: NCOMMS-23-36736A

Title: Improved Biomass Burning Emissions from 1750 to 2010 using Ice Core Records and Inverse Modeling

Author(s): Bingqing Zhang, Nathan J. Chellman, Jed O. Kaplan, Loretta J. Mickley, Takamitsu Ito, Xuan Wang, Sophia M. Wensman, Drake McCrimmon, Jorgen Peder Steffensen, Joseph R. McConnell, and Pengfei Liu

Response to Reviewer #1

Comment 1

The study mainly utilized inversion techniques to provide the BC emissions. It is recommended to modify the title "...emission" to "...emission of BC".

Response

Although we performed the inversion for BC, we aim to use it as a tracer to infer biomass burning emission changes, because the budget is simple. We finally derived biomass burning emissions for other gaseous and aerosol species by applying scale factors. Besides, our analysis is not limited to BC. We used the whole biomass burning dataset to estimate the change of cloud condensation nuclei from preindustrial period to the present day and calculate cloud albedo forcing. We also provided another example of the potential application of our dataset, which is estimating phosphorus emissions and depositions from biomass burning.

Comment 2

According to the response and the associated revision, the study could not provide the improved emissions at low latitudes. The study needs to provide further quantitative data to demonstrate that the emissions at mid- and high-latitudes account for the majority of global emissions. Otherwise, it does not show that the results of this study improve global emissions. The title should be revised to be consistent with the research results (e.g., Improved... emissions at mid- and high-latitudes from...).

Response

We think that the Illimani ice core record (in South America) shows constraint on the emissions in Arc of deforestation (ARCD), which is a low-latitude region. The emission difference in this region is smaller in the *a posteriori* emissions compared with *a priori* emissions (Figure S4). As mentioned in lines 91-103, we think emissions in regions in Figure S4 are relatively well constrained by the ice core record. The total biomass burning emission in these regions contributes to 40% - 60% of the global total and more than 90% of the emission differences between BB4CMIP and LPJ-LMfire (Figure S6). Our

inverse modeling can provide a tight constraint for emissions in these regions and reduce the discrepancies, thus improving the estimates of global total.

Comment 3

A priori inventories from different sources (e.g., biomass burning, fossil fuel/biofuel) were combined in the treatment of a priori inventories for inversion. However, the methodology described in the study could only obtain total rBC emissions and could not be able to distinguish the different emission sources of rBC, if the relationship between the different emission sources and the ice-core rBC records were not established separately. How did the study determine the emissions from Biomass Burning of rBC? Otherwise, it is recommended to modify the title to “Improved Emissions of BC at mid- and high-latitudes from...”.

Response

As we described in Lines 390-410, we treated emissions from biomass burning and fossil fuel/biofuel burning separately by setting different *a priori* emission errors. In short, we assumed errors for fossil fuel/biofuel emissions are much smaller than that for biomass burning emissions, as more records (such as energy production and consumption data, population record) are available for better estimation emissions from anthropogenic activities. The inverse modeling process made different adjustment on the *a priori* emissions from different sources based on the errors we set. The input of the inverse modeling (i.e., *a priori* emissions \mathbf{x}_A) is a 34 by 1 vector that represents the *a priori* BB emissions in 17 regions and fossil fuel/biofuel emissions in 17 regions, and the output (i.e., *a posteriori* emissions \mathbf{x}) is a 34 by 1 vector that represents the optimized estimates of BB emissions and fossil fuel/biofuel emissions.